# Fairness with respect to Stereotype Predictors: Impossibilities and Best Practices

**Inbal Livni Navon**                                                    *inballn@bgu.ac.il*
*Ben-Gurion University of the Negev*

**Omer Reingold**                                                        *reingold@stanford.edu*
*Stanford University*

**Judy Hanwen Shen**                                                     *jhshen@stanford.edu*
*Stanford University*

**Reviewed on OpenReview:** *https://openreview.net/forum?id=FPJKZDzdsW&*

## Abstract

As AI systems increasingly influence decision-making from consumer recommendations to educational opportunities, their accountability becomes paramount. This need for oversight has driven extensive research into algorithmic fairness, a body of work that has examined both allocative and representational harms. However, numerous works examining representational harms such as stereotypes encompass many different concepts measured by different criteria, yielding many, potentially conflicting, characterizations of harm. The abundance of measurement approaches makes the mitigation of stereotypes in downstream machine learning models highly challenging. Our work introduces and unifies a broad class of auditors through the framework of *stereotype predictors*. We map notions of fairness with respect to these predictors to existing notions of group fairness. We give guidance, with theoretical foundations, for selecting one or a set of stereotype predictors and provide algorithms for achieving fairness with respect to stereotype predictors under various fairness notions. We demonstrate the effectiveness of our algorithms with different stereotype predictors in two empirical case studies.

## 1 Introduction

Past works in algorithmic fairness have presented many perspectives on understanding harms perpetrated by machine learning models. One dominant line of work is in mitigating allocative harms by achieving group fairness, where the goal is to ensure individuals from different groups (or overlapping subgroups) do not experience disparate treatment (Hardt et al., 2016; Kim et al., 2019). Another category of harms, representational harms, arises when certain groups of people are stereotyped or stigmatized (Suresh & Guttag, 2021). For example, stereotypical associations, have been studied in computer science across many modalities including risk prediction (Wang & Russakovsky, 2021), natural language (Zhao et al., 2017), computer vision (Hall et al., 2022), and robotics (Hundt et al., 2022). Beyond computer science, the origins of "stereotypic thinking", holding simplified beliefs about members of specific groups, have been widely studied by social psychologists (Dovidio & Gaertner, 1993; Hilton & Von Hippel, 1996).

In computer science, many different definitions under a broad family of stereotypical associations have been proposed in empirical audit studies (Zhao et al., 2017; Wang & Russakovsky, 2021; Cheng et al., 2023a). For example, job classification algorithms may favor biographies written with words that are *typical* to the majority gender in the occupation (Cheng et al., 2023a). As another example, Wan et al. (2023) measure *harmful* gender stereotypes such as power, agency, and ability in LLM-generated recommendation letters. These two examples represent two distinctly different, yet valuable, concepts of stereotypes. Unlike

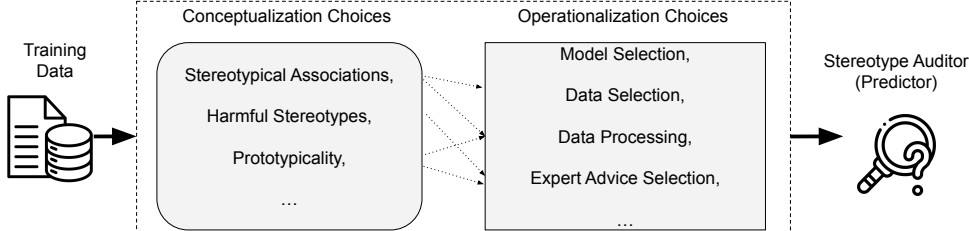

Figure 1: Existing research in measuring stereotypes encompasses a broad set of conceptualization and operationalization choices; leading to many potential downstream auditors of stereotypes (we term *stereotype predictors*). We propose a unifying framework of fairness with respect to these stereotype predictors that allows us to reason about how to compare and apply these predictors to mitigate stereotype-related harms.

the group fairness setting where protected populations (e.g. group labels) are known, there can be many different stereotypes for the same group that a decision-making algorithm (e.g. an automated hiring system) may want to minimize. The choices of which concept to measure and which model or data to choose yields numerous different auditors for stereotypes and other representational harms (Figure 1). Under the lens of measurement theory (Jacobs & Wallach, 2021), we can decompose the plethora of different approaches to measuring stereotypes into different notions of stereotypes (i.e. conceptualization choices) and different ways of auditing (i.e. operationalization choices).

Yet allocative and representational harms are not mutually exclusive. Representational harms may eventually lead to allocative harms when undesirable associations such as stereotypes proliferate the training data used for model development. Cheng et al. (2023a) measure the correlation between stereotypical associations in biographies and occupational predictors that are trained on the same biographies; the correlation they find reveals that words associated with gender may be stereotypically used to predict occupation.

Our goal is to bridge the measurement of stereotype-based representational harms with methods to prevent these stereotypes from evolving into allocative harms. For example, even if there exists an association between gender and occupational words in biographies, we want to design predictors that do not rely on gender associations to predict occupations. To this end, we need to consider the many potential constructions of stereotype measurements and consider how to be fair with respect to potentially all of them. Moreover, there exist gender stereotypes (e.g. college major) that are more closely associated with the outcome (e.g. occupation) than group membership. This demonstrates that choosing the stereotype and fairness interventions is a complex and consequential decision. In this work, we introduce a unifying framework for fairness with respect to stereotypes.

## 1.1 Our Contributions

**Fairness Definition with Respect to Stereotypes:** We define stereotype predictors and relate them to the existing literature on measuring continuous fairness notions and proxy predictors. We then propose covariance-based versions of demographic parity, equalized opportunity, and multiaccuracy with respect to stereotypes.

**A Guide to Choosing Stereotype Predictors:** For each group, there may exist a multitude of different stereotype predictors. We address the key questions to achieving fairness with respect to these predictors:

- *Which stereotype predictors should be used?* Enforcing fairness with respect to some stereotype predictors comes at the cost of the model's accuracy. For demographic parity and equal opportunity, we show when a stereotype predictor is correlated with the outcome, model accuracy is reduced.

- *Can we achieve fairness with respect to multiple stereotype predictors?* Since multiple stereotype predictors can arise from different concerns, model choices, and data selection techniques, we investigate if fairness with respect to multiple stereotypes can be achieved simultaneously. We show that in some cases only a constant predictor can satisfy demographic parity or equal opportunity with

respect to a few different stereotypes. However, for multiaccuracy, fairness with respect to multiple stereotypes can be achieved.

- *Are different notions of fairness with respect to stereotypes compatible?* We show when multiaccuracy and demographic parity cannot be satisfied together. We also extend the result from Kleinberg et al. (2016) and show that in all non-trivial cases, we cannot satisfy multiaccuracy and equalized odds with respect to stereotypes.

**Post-Processing Algorithms and Empirical Case Studies**   We give simple post-processing algorithms for demographic parity and equal opportunity with respect to stereotype predictors. We present two case studies in the areas of professional biographies and educational surveys to demonstrate the effectiveness of our algorithms. For each case study, we create three plausible stereotype predictors based on different conceptual and operational decisions. We demonstrate the effectiveness of our simple post-processing algorithms in achieving fairness with respect to each of the stereotype predictors as well as the ensemble of all of them.

Our work highlights the challenges of pursuing fairness with respect to stereotypes that are different from achieving fairness guarantees for protected groups. Specifically, we find that stereotype-based notions might incur a higher cost on accuracy than group-based notions. Since multiple stereotypes could apply to the same group, particular care should be taken in determining which stereotypes to protect against.

## 2    Related Work

**Group Fairness**   The formalization of fairness notions pertaining to groups has generated different fairness definitions for different scenarios (Hardt et al., 2016; Kim et al., 2019). These definitions cannot be simultaneously achieved unless the underlying data has certain properties (e.g. equal base rates) (Kleinberg et al., 2016; Kim et al., 2019; Hébert-Johnson et al., 2018). Prior works have also proposed techniques to consider continuous variables maximal correlation (Mary et al., 2019; Grari et al., 2019). Our work does not require such strong independence assumptions and focuses on post-processing methods rather than in-processing methods. For demographic parity, continuous definitions have been proposed for specific features such as race ratio (Jiang et al., 2022).

**Stereotypes and Social Norm Bias**   Many recent works have identified that enforcing constraints based on group membership alone is not sufficient to guarantee equitable outcomes and may harm individuals if unintended discrimination occurs within a group (Lipton et al., 2018; Cheng et al., 2023a). Inferred group norms, or over-generalized beliefs of certain groups, are also referred to as *stereotypes* (Nadeem et al., 2020). For example, words that are associated with one group more often than other groups may appear in model outputs in tasks such as language generation (Nadeem et al., 2020), semantic role labeling (Zhao et al., 2017), and professional biography classification (Cheng et al., 2023a). Works in this area develop methods to measure stereotypes (Cryan et al., 2020; Sánchez-Junquera et al., 2021; Seaborn et al., 2023; Bosco et al., 2023) and audit for the presence of stereotypes (Kambhatla et al., 2022; Cheng et al., 2023b;a).

**Proxy Fairness**   When sensitive attributes are not available due to missing data or privacy concerns, proxy predictors have been proposed (Chen et al., 2019; Diana et al., 2022; Zhu et al., 2023). Standard fairness definitions can then be enforced according to these proxy predictors. Training a proxy predictor to ensure fairness requires explicitly learning the association between features and group membership. Stereotypes compass a wider range of social norms that may not be directly indicative of group membership (e.g. hair length, the presence of the word "activism" in a biography) (Lipton et al., 2018; Cheng et al., 2023a).

## 3    Preliminaries

We denote by $\mathcal{X}$ the set of individuals, and assume that there is a binary outcome $y \in \{0, 1\}$ that we want to predict, and a distribution $(x, y) \sim \mathcal{D}$ over the individuals. In our example, $y = 1$ implies that the individual belongs to the positive prediction class. Let $g : \mathcal{X} \to \{0, 1\}$ be an indicator function for the protected group.

All expectations, variances, covariances, and correlations in our paper are defined over $\mathcal{D}$. A predictor for $y$ is a function $p : \mathcal{X} \to [0,1]$. In some cases, we require the predictor to have a discrete range $R \subset [0,1]$.

We state some standard group fairness definitions of a predictor $p$ with respect to a group $g$, with slackness parameter $\alpha \in [0,1]$. We remark that multiaccuracy can also be defined with respect to a continuous $g$.

**Definition 3.1.** *A predictor* $p : \mathcal{X} \to [0,1]$ *satisfies* $\alpha$*-demographic parity with respect to a group $g$ if* $\left| \mathbb{E}_{(x,y) \sim \mathcal{D}}[p(x)|g(x) = 1] - \mathbb{E}_{(x,y) \sim \mathcal{D}}[p(x)|g(x) = 0] \right| \leq \alpha$.

**Definition 3.2.** *A predictor* $p : \mathcal{X} \to [0,1]$ *satisfies* $\alpha$*-equal opportunity with respect to a group $g$ if* $\left| \mathbb{E}_{(x,y) \sim \mathcal{D}}[p(x)|g(x) = 1, y = 1] - \mathbb{E}_{(x,y) \sim \mathcal{D}}[p(x)|g(x) = 0, y = 1] \right| \leq \alpha$.

**Definition 3.3.** *A predictor* $p : \mathcal{X} \to [0,1]$ *satisfies* $\alpha$*-multiaccuracy with respect to a set of groups $S$ if for all $g \in S$,* $\left| \mathbb{E}_{(x,y) \sim \mathcal{D}}[g(x)(y - p(x))] \right| \leq \alpha$.

# 4 Auditing for Stereotypes: A Unified Perspective

The fairness definitions we have highlighted thus far are based on known and well-defined group membership. These definitions allow us to guarantee, for example, that an equal number of men and women are being interviewed for the job (i.e. demographic parity). However, many works have pointed out additional harms that remain even when group fairness constraints are satisfied. For example, Cheng et al. (2023a) identify stereotypical associations to be harmful to members of minority groups who may have more stereotypical features; this could lead to some stereotypically presenting individuals being unjustly denied opportunities even though group notions of fairness are satisfied. More broadly, stereotypical associations fall under the category of representational harms, where certain people or groups are stereotyped or stigmatized (Suresh & Guttag, 2021; Katzman et al., 2023; Chien & Danks, 2024). To capture the broader notions of stereotypes under representational harms, we need to map an individual's features to their association with a group that is flexible enough to capture different concepts of stereotypes. We designate such a mapping as a *stereotype predictor*. While widely held stereotypical beliefs can be inaccurate (Alexander, 1992; Beeghly, 2021), in this work we are concerned about stereotypes that a machine learning algorithm learns from the data, and therefore restrict our attention to stereotypes that are positively correlated with the group membership.

**Definition 4.1.** ***Stereotype Predictor*** *Given a set of individuals $\mathcal{X}$ with group membership $g : \mathcal{X} \to \{0,1\}$, a stereotype predictor $p_g : \mathcal{X} \to [0,1]$ is a function describing the degree to which $x$ belongs to group $g$, such that $\Pr_{x \sim \mathcal{D}}[g(x) = 1 | p_g(x) = r]$ is monotonically increasing with $r$.*

While we expect a stereotype $p_g$ for the group $g$ to correlate with the group membership, we do not require it to be calibrated. Our technical results hold even in cases where stereotypes show no statistical correlation with the group membership, though such scenarios are uncommon. Our definition of a stereotype predictor is flexible enough to capture many notions presented in prior literature, the following are three examples:[1]

- **Stereotypical Associations**: Given all features of an individual, stereotypical associations arise when an individual is very closely associated with their group. Thus, the stereotype predictor for group associations, $p_g$, uses *all* features of a user to predict group membership. Auditing the decision-making model for fairness with respect to this $p_g$ can reveal how much of the model decision is made based on group associations. Social Norm Bias (Cheng et al., 2023a) is an example of this auditing approach. If $p_g$ were further restricted to binary outputs (i.e. $p_g : \mathcal{X} \to \{0,1\}$), notions of proxy fairness also fall within this category (Chen et al., 2019; Diana et al., 2022; Zhu et al., 2023).

  Procedurally, the main approach for finding such $p_g$ is using a learning algorithm on labeled data pairs of features and group labels $D = \{(x,g)\}_{i=1}^{n}$ (i.e. $p_g = \arg\min_{p_g} \mathbb{E}_{(x,g) \in D}[l(p_g(x), g)]$ for some loss $l$). Different choices around datasets and model classes will result in many different group association stereotype predictors being produced.

- **Harmful Stereotypes**: Concepts or words that perpetuate harmful perceptions of certain groups are considered harmful stereotypes. In a thorough study of measuring stereotypes in language, Blodgett et al. (2021) highlight that the construct of a stereotype requires the co-occurrence of

---

[1]See Appendix A for a discussion on these different types of stereotype predictors.

the group membership and features that may result in harm. Using specific features is a common approach for auditing generative models (Cryan et al., 2020; Sánchez-Junquera et al., 2021; Seaborn et al., 2023; Bosco et al., 2023). Moreover, some stereotypes are not only harmful but also reflect existing injustices in society (Jussim et al., 2009).

For this notion, harmful stereotypes are often identified through human annotations (Kambhatla et al., 2022; Nangia et al., 2020). When taking multiple harmful features, we can combine them into a single auditor predictor using the predictor described above when applying it only on the harmful features, or using the clustering techniques described next. Decisions around which harmful stereotypes to include and how harmful stereotypes are annotated will result in different stereotype predictors that share this same concept.

- **Prototypical vs Outlier Individuals**: In Dwork et al. (2012), the authors introduced the notion of individual fairness. In individual fairness, we assume that there is a distance metric between the individuals, and require that close individuals in this metric are treated similarly. Given a distance metric between the individuals, we can extend the idea of individual fairness to the notion of the level of conformity to the group. Using the distance to the group mean yields a stereotype predictor that audits whether similar individuals are treated similarly even if they belong to different groups and whether prototypical individuals are treated differently than outliers in groups.

  A stereotype predictor $p_g$ based on a distance metric can be implemented using algorithms for Fuzzy clustering (Miyamoto et al., 2008). Fuzzy clustering algorithms create clusters and give each individual a weight vector representing how likely it belongs to each of the clusters. The importance of clustering that is robust and fair to outliers has been highlighted in prior work (Charikar et al., 2001; Almanza et al., 2022).

### 4.1 Stereotypes Through Covariance

We have defined the main quantities of interest: the true labels ($y$), the true group membership ($g$), the predictor for a task of interest ($p$), and a stereotype predictor ($p_g$) that represents a stereotype we wish to be agnostic to. Prior work formalizing fairness notions provides the scaffolding to understand what types of disadvantages can be measured and how different notions relate to and preclude one another (Hardt et al., 2016; Kleinberg et al., 2016). We now formalize notions of fairness with respect to stereotype predictors.

We present these definitions below as extensions of the group fairness definitions for a stereotype predictor. For a stereotype predictor with a binary output, $p_g : \mathcal{X} \to \{0, 1\}$, definitions 4.2 and 4.3 below are equivalent to the standard group fairness definitions up to the error parameter. For multiaccuracy, the dependency is slightly more complex [2].

**Definition 4.2** (Demographic Parity With Respect To Stereotypes). *A predictor $p : \mathcal{X} \to [0, 1]$ satisfies $\alpha$-demographic parity with respect to a stereotype predictor $p_g$ if $|\mathrm{Cov}(p, p_g)| \leq \alpha$.*

**Definition 4.3** (Equal Opportunity With Respect To Stereotypes). *A predictor $p : \mathcal{X} \to [0, 1]$ satisfies $\alpha$-equal opportunity with respect to a stereotype predictor $p_g$ if $|\mathrm{Cov}(p, p_g|y = 1)| \leq \alpha$.*

**Definition 4.4** (Multiaccuracy With Respect To Stereotypes). *Let $\alpha \geq 0$, a predictor $p : \mathcal{X} \to [0, 1]$ satisfies $\alpha$-covariance multiaccuracy with respect to a set $S$ of stereotype predictors if for all $p_g \in S$, $|\mathrm{Cov}(p, p_g) - \mathrm{Cov}(y, p_g)| \leq \alpha$.*

Some of our results also apply for other group fairness definitions such as equalized odds and multicalibration. In addition, all of the above definitions can also be defined when conditioning on $g = 1$ and $g = 0$, as in Cheng et al. (2023a)[3]. These definitions are adjacent to bias amplification observed in prior work on stereotypes (Zhao et al., 2017; Wang & Russakovsky, 2021)[4].

Using these quantities, we measure the stereotypical behavior using the covariance between the predictor $p$ and $p_g$. We chose covariance over linear (i.e. Pearson) correlation for two reasons: (1) Preciseness:

---

[2]See Appendix C.1 for the exact statements.
[3]The additional definitions appear on Appendix C.2
[4]See Appendix F for a discussion on bias amplification and its relation to multiaccuracy.

Covariance expresses the true relationship between the values output by $p$ and $p_g$ without normalization. This is important because we assume the output of $p$ can be directly used for decision-making. (2) Robustness: In cases where $p$ (or $p_g$) has a small variance, very small changes can have a large effect on the correlation $\rho(p, p_g)$, while not changing the output we care about.

# 5   A Guide to Choosing Stereotype Predictors

Now that we have presented the analogous notions of fairness with respect to stereotypes, the natural question arises: *When and how should we use them?* Prior works have illustrated that tradeoffs arise between different notions of fairness with respect to group membership and in this section we extend some of these results to fairness with respect to stereotypes.

Our results show that enforcing fairness notions with respect to stereotypes is even more complicated and nuanced. In this case, the choice is not only which fairness definition to use, but also which stereotype predictors $p_g$ to apply since multiple exist. The existence of multiple possible stereotype predictors ($p_g$) is a challenge highlighted by prior work (Cheng et al., 2023a; Zhu et al., 2023). Unlike the setting of group fairness, where group membership is well-defined, there are many different stereotypes for the same group. Some fairness definitions can handle many different stereotypes, while others cannot. In this section, we investigate different fairness definitions with respect to different stereotypes, in order to help guide practitioners.

## 5.1   Which Stereotype Predictors Should be Used?

When picking a stereotype predictor to be used for achieving demographic parity and equal opportunity, the relationship between $p_g$ and the outcome $y$ must be carefully considered. We show that for demographic parity and equal opportunity, choosing a $p_g$ that is too correlated with the outcome will lower a predictor's accuracy while multiaccuracy is not sensitive to this relationship.[5]

**Lemma 5.1.** *For every $0 \leq \alpha < \gamma \leq 1$, let $p_g$ be a predictor such that $\mathrm{Cov}(y, p_g) > \gamma$, and let $p^* : \mathcal{X} \to [0, 1]$ be the distribution of $y$ given $x$. Then any predictor $p : \mathcal{X} \to [0, 1]$ satisfying $\mathrm{Cov}(p, p_g) \leq \alpha$ also satisfy: $\mathbb{E}_{(x,y) \sim \mathcal{D}}[(p(x) - y)^2] \geq \mathbb{E}_{(x,y) \sim \mathcal{D}}[(p^*(x) - y)^2] + 4(\gamma - \alpha)^2$.*

When picking a stereotype predictor with a very small $\mathrm{Cov}(y, p_g)$, we capture the irrelevant stereotypes - those that have no connection to the outcome. Enforcing demographic parity with respect to irrelevant stereotypes is equivalent to multiaccuracy. Since stereotype predictors can be chosen, there exists predictors that are more correlated with the outcome than the group membership itself. In this case, requiring demographic parity with respect to these predictors harms the accuracy significantly more than demographic parity with respect to group membership. [6]

Next, we prove a similar statement for equal opportunity.

**Lemma 5.2.** *Let $(x, y) \sim \mathcal{D}$ and denote by $p^* : \mathcal{X} \to [0, 1]$ the distribution of $y$ given $x$. For every $0 \leq \alpha < \gamma \leq 1$, let $p_g : \mathcal{X} \to [0, 1]$ be a stereotype predictor such that $\mathrm{Cov}(p^*, p_g | y = 1) > \gamma$. Then, any predictor $p$ satisfying $\alpha$-equal opportunity with respect to $p_g$ also satisfies $\mathbb{E}_{(x,y) \sim \mathcal{D}}[(p(x) - y)^2] \geq \mathbb{E}_{(x,y) \sim \mathcal{D}}[(p^*(x) - y)^2] + (\gamma - \alpha)^2 \left( \mathbb{E}_{(x,y) \sim \mathcal{D}}[y] \right)^2$.*

Unlike Lemma 5.1, where we can check if $\mathrm{Cov}(y, p_g) > \gamma$, the condition in Lemma 5.2, $\mathrm{Cov}(p^*, p_g | y = 1) \geq \gamma$, cannot be estimated from the data. Samples of $(x, y) \sim \mathcal{D}$ do not suffice for approximating $\mathbb{E}[(p^*(x))^2]$, which is required for the approximating the covariance in Lemma 5.2. This lemma shows the *existence* of a setting where enforcing equal opportunity is harmful to the accuracy. We also give a direct statement, of when post-processing a predictor to satisfy equal opportunity will reduce its accuracy. The optimal post-processing of $p$ is the best function $f(p(x), p_g(x)) : \mathcal{X} \to [0, 1]$ that minimizes a given loss, see Section 6.4. Both lemmas also apply to equalized odds, which is a restriction of equal opportunity[7].

---

[5] All proofs appear in Appendix D.1.
[6] See Appendix A.1 for a synthetic data demonstration.
[7] See Appendix C.2 for the definition.

**Lemma 5.3.** *Let $p, p_g : \mathcal{X} \to R$ be two predictors with discrete range. Let $p_{opt} : \mathcal{X} \to [0, 1]$ be the optimal post-processing of $p$ that minimizes the $\ell_2$ loss without constraints. Let $p'$ be the optimal post-processing of $p$ with $\ell_2$ loss that satisfies $\alpha$-equal opportunity with respect to $p_g$. If $\mathrm{Cov}(p_{opt}, p_g) > \gamma$ and $\mathbb{E}_{(x,y)\sim\mathcal{D}}[(p_{opt}(x) - y)^2] \leq \frac{1}{2}(\gamma - \alpha)^2 \left(\mathbb{E}_{(x,y)\sim\mathcal{D}}[y]\right)^2$, then $\mathbb{E}_{(x,y)\sim\mathcal{D}}[(p'(x) - y)^2] \geq \frac{1}{2}(\gamma - \alpha)^2 \left(\mathbb{E}_{(x,y)\sim\mathcal{D}}[y]\right)^2$.*

When requiring covariance multiaccuracy, we require $p$ to be similar to the outcome $y$. Requiring multiaccuracy does not reduce the accuracy of the predictor. In Lemma 6.1 we show that the covariance multiaccuracy is derived from the standard multiaccuracy, and in Lemma 7.6 in Gopalan et al. (2023) (originally in Hébert-Johnson et al. (2018)), they show that requiring multiaccuracy improves the accuracy of the predictor.

> For Demographic Parity (Lemma 5.1) and Equal Opportunity (Lemma 5.2) with respect to stereotype predictors ($p_g$), choose a $p_g$ that is not closely associated with the outcome ($y$). Any $p_g$ works for multiaccuracy, since multiaccuracy is equivalent to no bias amplification (Claim 6.1).

### 5.2 Can We Achieve Fairness with Respect to Multiple Stereotype Predictors?

Multiple different stereotype predictors can be estimated for protected groups but achieving fairness with respect to all stereotype predictors is not guaranteed to preserve accuracy for demographic parity and equalized odds. We show using an example, that requiring demographic parity or equal opportunity with respect to multiple stereotype predictors risks forcing the predictor to become a constant predictor.[8]

**Lemma 5.4.** *There exists a simple distribution $\mathcal{D}$ over $\mathcal{X} \times \{0, 1\}$, a protected group $g \subset \mathcal{X}$ and a set of stereotype predictors $p_g$ for $g$ such that any predictor $p$ satisfying $\alpha$-demographic parity with respect to all $p_g$ is approximately the constant predictor.*

**Lemma 5.5.** *There exists a simple distribution $\mathcal{D}$ over $\mathcal{X} \times \{0, 1\}$, a protected group $g \subset \mathcal{X}$ and a set of stereotype predictors $p_g$ for $g$ such that such that any predictor $p$ satisfying $\alpha$-equal opportunity with respect to all $p_g$ is approximately the constant predictor.*

The main idea in the proofs is to construct a set of predictors $p_g$ that are all calibrated, but very different. Requiring demographic parity or equal opportunity forces the predictor $p$ to treat different sets of individuals the same, and combining many such requirements results in $p$ treating all individuals the same. We remark that although the proofs are constructed with a few stereotype predictors over a simple distribution, a similar phenomenon can happen on real datasets as the number of different stereotype predictors increases.

**Comparison with Multiaccuracy** By enforcing any fairness requirement with respect $p_g$ on $p$, we introduce a constraint on $p$. This is true for demographic parity, equal opportunity, and multiaccuracy type constraints. In the case of multiaccuracy, the constraints are those that $p^*$ satisfies. Therefore, when enforcing multiaccuracy with respect to many different stereotypes, the prediction of $p$ improves and becomes closer to $p^*$. In the case of demographic parity and equal opportunity, the constant predictor always satisfies the constraints, and using many different stereotype predictors $p_g$ pushes $p$ towards being constant.

> There are some combinations of stereotype predictors where achieving Demographic Parity (Lemma 5.4) or Equal Opportunity (Lemma 5.5) with respect to all stereotype predictors results in a constant predictor. Choose sets of stereotype predictors that do not contradict if fairness with respect to multiple stereotypes is desired.

### 5.3 Are Different Notions of Fairness with Respect to Stereotypes Compatible?

We show very simple settings in which no predictor can satisfy both demographic parity and multiaccuracy with respect to a stereotype $p_g$. Our results complement existing results in the standard group setting, where different fairness notions are mostly not compatible (Kleinberg et al., 2016).[9]

---

[8]All proofs appear in Appendix D.2

[9]The proof appears on Appendix D.3.

**Lemma 5.6.** *There exists a simple distribution $\mathcal{D}$ over $\mathcal{X} \times \{0,1\}$, a protected group $g \subset \mathcal{X}$ and a two sets of stereotype predictors $S_1, S_2$ for $g$ such that no predictor $p$ can satisfy both multiaccuracy with respect to $p_g \in S_1$ and demographic parity with respect to $p_g \in S_2$.*

**Impossibility Theorem for Stereotype Predictors** Hardt et al. (2016) introduced both equal opportunity and a stronger measure, equalized odds, where both the false positive error and the false negative error must be equal between the groups. Kleinberg et al. (2016) showed that it is not possible to achieve both calibration and equalized odds. In this paper, we prove the corresponding result with respect to stereotype predictors.[10]

**Theorem 5.7.** *Let $\mathcal{D} \subset \mathcal{X} \times \{0,1\}$ be a distribution. Assume that every predictor $p' : \mathcal{X} \to [0,1]$ has $\operatorname{Cov}(y, p') \leq \frac{9}{10}(\mathbb{E}_{(x,y)\sim\mathcal{D}}[y] - \mathbb{E}^2_{(x,y)\sim\mathcal{D}}[y])$. Let $p_g : \mathcal{X} \to [0,1]$ be a stereotype that is correlated with the outcome: $\operatorname{Cov}(y, p_g) \geq \beta$. Let $\alpha < \frac{1}{30}\beta$ be a parameter. Then, no predictor $p : \mathcal{X} \to [0,1]$ can satisfy both of the following two properties:*

1. *$\alpha$-multiaccuracy with respect to $p_g$: $\left|\mathbb{E}_{(x,y)\sim\mathcal{D}}[p_g(x)(y - p(x))]\right| \leq \alpha$ and the expectation is calibrated $\left|\mathbb{E}_{(x,y)\sim\mathcal{D}}[p(x)] - \mathbb{E}_{(x,y)\sim\mathcal{D}}[y]\right| \leq \alpha$.*

2. *Equalized odds with respect to $p_g$: for $b \in \{0,1\}$, $\operatorname{Cov}(p, p_g|y = b) \leq \alpha$.*

The condition $\operatorname{Cov}(y, p') \leq \frac{9}{10}(\mathbb{E}_{(x,y)\sim\mathcal{D}}[y] - \mathbb{E}^2_{(x,y)\sim\mathcal{D}}[y])$ for every predictor $p'$ implies that the outcome $y$ is unpredictable to some extent. In the case of a known deterministic $y$, it is possible to satisfy both definitions.

> Some stereotype predictors make achieving multiple notions of fairness impossible. Check a stereotype predictor, $p_g$ for correlation with the outcome $y$, before applying multiple fairness post-processing algorithms.

## 6 Post-Processing for Different Fairness Notions

In this section, we describe efficient post-processing algorithms that satisfy different fairness notions with respect to stereotypes [11].

### 6.1 Demographic Parity

Given a predictor $p$ and a set $S$ of stereotype predictors $p_g$, we show an efficient post-processing of $p$ to satisfy demographic parity with respect to all $p_g \in S$. The `clip` function that clips the value to the range $[0,1]$. The above process converges after at most $1/\alpha^2$ iterations. The goal of the algorithm is to output a predictor that is close to the original $p$ and satisfies demographic parity with respect to all stereotype predictors $p_g$.

---

**Algorithm 1:** Demographic Parity Post Processing

---

**while** $\exists p_g \in S$ *such that* $|\operatorname{Cov}(p, p_g)| > \alpha$ **do**
    $b := \operatorname{sign}(\operatorname{Cov}(p, p_g))$
    $p(x) \leftarrow \texttt{clip}(p(x) - b\alpha \cdot p_g(x)$

---

### 6.2 Equal Opportunity with Respect to a Single Stereotype Predictor

Given a stereotype predictor $p_g : \mathcal{X} \to R$ with a discrete range $R$, and a candidate predictor $p$, we suggest a simple post-processing of $p$ to produce $p'$ that satisfies equal opportunity with respect to $p_g$. The algorithm takes the set of inputs $x$ for which $p_g(x) > \mathbb{E}[p_g(x)|y = 1]$, and reduces $p(x)$ on this set. This simple transformation is a uniform change to $p(x)$ which achieves equal opportunity.

---

[10]The definitions appears on Appendix C.2 , and the proof on Appendix D.4
[11]The analysis of the algorithms can be found in Appendix E.1 (Algorithm 1) and Appendix E.2 (Algorithm 2).

---

**Algorithm 2:** Equal Opportunity Post Processing

---

$\mu_g := \mathbb{E}_{(x,y) \sim \mathcal{D}}[p_g(x)|y = 1]$

$R' := \{r \in R \mid r > \mu_g\}$

$p_1(x) := p(x) \cdot \mathbf{1}(p_g(x) \in R')$ and $\forall x$ , $p_2(x) := p(x) \cdot \mathbf{1}(p_g(x) \notin R')$ $\forall x$

Set $\gamma := -\mathrm{Cov}(p_2, p_g|y = 1)/\mathrm{Cov}(p_1, p_g|y = 1)$

$p'(x) := \begin{cases} \gamma p(x) & p_g(x) \in R' \\ p(x) & \text{otherwise} \end{cases}$

---

The same post-processing algorithm with a small modification can also be used to achieve demographic parity. In order to do so, we apply the same algorithm only we remove the condition of $y = 1$ from both the expectation and the covariance. When we are interested in equal opportunity with respect to individuals in which $y = 1$ and $g = 1$, as in Cheng et al. (2023a), we can use this algorithm when also conditioning on $g = 1$.

### 6.3 Multi-Accuracy

In the following lemma, Lemma 6.1[12], we show that a predictor $p$ that satisfies $\alpha$ covariance multiaccuracy with respect to a stereotype predictor $p_g$, satisfies the multiaccuracy condition with parameter $2\alpha$. Therefore, we can post-process a predictor $p$ by using the multiaccuracy algorithm with an error parameter $\alpha/2$ (see Section 7 in Gopalan et al. (2023) for the algorithm). This algorithm can handle a set of continuous functions, so we can use it to achieve multiaccuracy with respect to a set of different stereotype predictors.

**Lemma 6.1.** *A predictor $p$ that is $\alpha$-multiaccurate with respect to $\{p_g, 1\}$ satisfies the covariance multiaccuracy definition with respect to $p_g$ with error $2\alpha$, $|\mathrm{Cov}(p, p_g) - \mathrm{Cov}(y, p_g)| \leq 2\alpha$.*

### 6.4 Optimal Post-Processing for all Fairness Definitions

Given a predictor $p : \mathcal{X} \to R$ with a discrete range $R$ and a small set of stereotype predictors $\{p_{g_i}\}_{i \in I}$, $p_{g_i} : \mathcal{X} \to R$, we formulate an optimization problem with linear constraints find the optimal post-processing that satisfies all constraints. This approach takes exponential time in $|I|$, and therefore might be slower than the previous post-processing algorithms for many parameter regimes.

As an overview of our approach, we denote the post-processed predictor by $p'$. For every value $r \in R^{|I|+1}$ we define a variable $z_r$ representing $p'(x)$ on all $x$'s such that $p_{g_i}(x) = r_i$ for all $i \in I$ and $p(x) = r_{|I|+1}$. Given a convex Lipschitz bounded loss function $\ell : R \times \{0, 1\} \to [0, 1]$, we create an optimization problem for the values of $z_r \in [0, 1]$ that minimizes the loss, while satisfying all of the constraints (if such solution exists).[13]

## 7 Experiments

### 7.1 Case Study: Gender Stereotypes in Biographies

Our first case study builds on our running example of gender stereotypes around professional language. [14] We examine a set of real-world professional biographies scraped from Common Crawl (Cheng et al., 2023a). We followed the same scraping process as De-Arteaga et al. (2019) and collected 46,878 biographies. For consistency with prior work, we also use word embeddings from FastText (Bojanowski et al., 2016) (WE) as feature representations and use a logistic regression classifier for fitting the occupational predictor and stereotype predictors (Cheng et al., 2023a). We designate the main task as a binary prediction task to predict whether the biography belongs to a lawyer.

---

[12]The proof appears in Appendix E.3.

[13]See Appendix E.4 for the explicit optimization problem.

[14]Code and data for all experiments are available here: `https://github.com/heyyjudes/formalizing-fairness-for-stereotypes`

**Auditing with Stereotype Predictors** There are conceptual and operational choices that could result in many different stereotype predictors. We examine three stereotype predictors for gender in this particular context:

- Biography Stereotypical Associations $\hat{p}_g^{(all)}$: We fit a gender predictor over all of the biographies in our dataset. Fairness with respect to this predictor guarantees the occupation prediction model does not depend on words stereotypically used across *all* biographies associated with each gender.

- Law Stereotypical Associations $\hat{p}_g^{(law)}$: We fit a gender predictor law biographies in our dataset. Fairness with respect to this predictor guarantees the occupation prediction model does not depend on words stereotypically used in *law* biographies associated with each gender.

- Law Prototypicality $\hat{p}_g^{(FCM)}$: We use a fuzzy clustering objective (Fuzzy c-means (Bezdek et al., 1984)) to measure how typical each biography is using the distance to the cluster center for each gender group according to the lawyer biographies clusters. Fairness with respect to this predictor guarantees the occupation prediction model does not depend on how typical the biography is to the center of each gender cluster.

For all three predictors, we use word embeddings (FastText Bojanowski et al. (2016)) to generate an averaged representation for each biography. For the stereotype association predictors, the logistic regression model, trained on embedding features, achieved 66% accuracy with a Pearson correlation coefficient ($\rho$) of approximately 0.10 for both predictors. The covariance between true labels and stereotype predictor ($Cov(y, \hat{p}_g^{(all)})$ and $Cov(y, \hat{p}_g^{(law)})$)) ranged from 0.010 to 0.015. Both association predictors showed statistically significant correlations with the target label. For the fuzzy c-means cluster predictor, we use findings from prior work (Bolukbasi et al., 2016) that there are gender subspaces in word embeddings and we identify the top 4 components of the embeddings that change the most across male and female words. We use these 4 dimensions for the fuzzy c-means predictor. The predictor achieves approximately 57% accuracy in gender prediction, with a near-zero covariance ($Cov(y, \hat{p}_g^{(FCM)}) = 0.002$) but a statistically significant correlation ($\rho = 0.10, p < 1e^{-4}$) between the true label and stereotype predictor. All three of these predictors are not perfectly predictive of gender and exhibit some correlation with the ground truth label. We will compare the effects of enforcing different fairness constraints with respect to these predictors.

**Fairness with respect to Stereotype Predictors**

For each stereotype predictor, we can enforce demographic parity according to the predictor and observe the effects on accuracy and disparity as a result. Figure 2a shows the results from using Demographic Parity Post Processing (Algorithm 1). For each of the stereotype predictors, (e.g. $\hat{p}_g^{(all)}$, $\hat{p}_g^{(law)}$, and $\hat{p}_g^{(FCM)}$), applying the post-processing algorithm successfully reduced $Cov(p, p_g)$ for the target $p_g$ without worsening Demographic Parity (DP) with respect to the other predictors. Applying DP to all three predictors successfully achieved significantly reduced $Cov(p, p_g)$ with respect to all stereotype predictors at the cost of a slight decrease in accuracy (88% to 85%). Figure 2b shows the results for applying Equal Opportunity Post Processing (Algorithm 2) for each respective stereotype predictor as well as the combination of all predictors. Reducing equal opportunity violation for $\hat{p}_g^{(FCM)}$ resulted in an increased violation with respect to the other stereotype predictors. Consequently, when we applied the algorithm to all three predictors stereotype together, $Cov(p, \hat{p}_g^{(law)}|y = 1)$ and $Cov(p, \hat{p}_g^{(all)}|y = 1)$ were much larger than individual post-processing with respect to $\hat{p}_g^{(law)}$ or $\hat{p}_g^{(all)}$. This is a case where we observe contradictions between stereotype predictors. A practitioner in this scenario should try a different prototypical predictor or just choose one concept of stereotypes (e.g. stereotype associations) to correct for.

## 7.2 Case Study: Racial Stereotypes in College Success

The next case study focuses on predicting educational attainment from a longitudinal survey. The National Educational Longitudinal Study is a nationally representative, longitudinal study of 8th graders in 1988 and follows students through their secondary and post-secondary educational attainment (e.g. until 2000). The purpose of this study is to understand the role of a comprehensive set of student backgrounds across

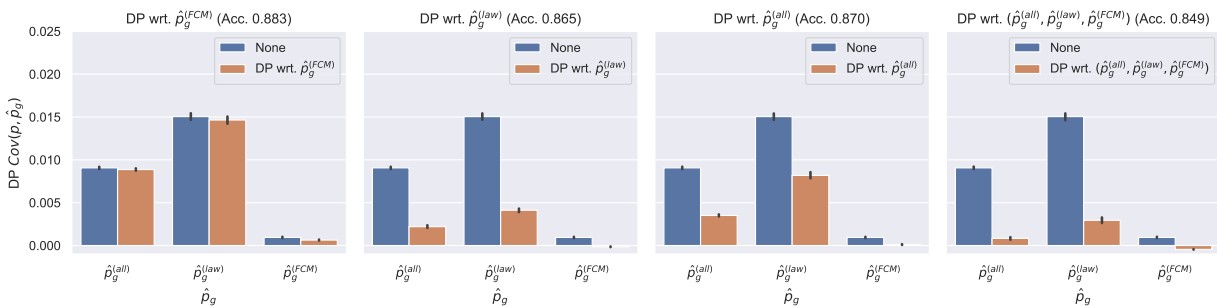

(a) Demographic Parity Post-Processing of an occupation predictor (law profession) with different stereotype predictors for gender. While applying all three stereotype predictors was effective, overall model accuracy was reduced by 3%.

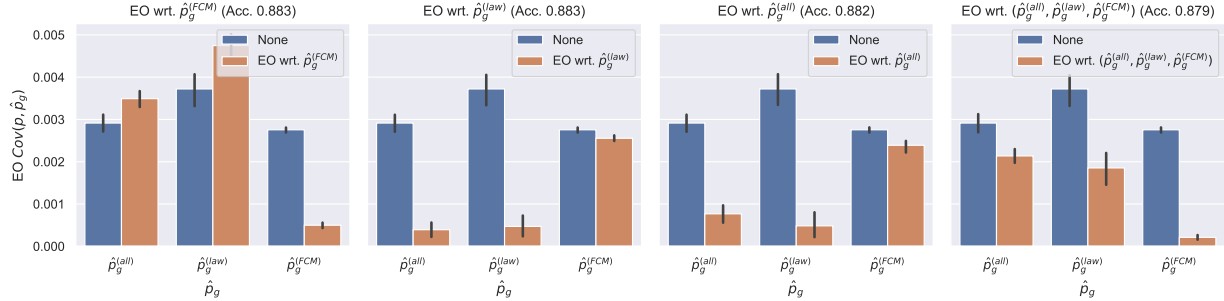

(b) Equal Opportunity Post-Processing of an occupation predictor (law profession) with different stereotype predictors for gender. Applying post-processing with respect to each stereotype predictor individually yielded better results than combining all of them.

Figure 2: Post-processing results for different fairness metrics

school and home experiences. This dataset is used by seminal work on algorithmic fairness by Kleinberg et al. (2018) in the setting where algorithmic predictions guide decisions. Specifically, they use this dataset to critically examine how including race improved the performance of predicting student performance in college. Predicting college success has been studied extensively in recent years (Jiang & Pardos, 2021; Yu et al., 2020; 2021; Kizilcec & Lee, 2022). Using this dataset, our goal is to audit and correct for stereotypes in predicting the attainment of obtaining a bachelor's degree. After the binarization of categorical variables and the removal of race and outcome variables, 812 features are then used for the task of prediction. We study the logistic regression models after standardizing features. We include only the entries for white ($n = 7908$) and black ($n = 1041$) students; together these two groups make up 74% of the entire dataset.

**Auditing with Stereotype Predictors** In this setting, we again have different conceptual and operational choices that could result in many different stereotype predictors. We examine three stereotype predictors for race:

- Stereotypical Associations $\hat{p}_g^{(all)}$: We fit a stereotype predictor over all student features in our dataset. Fairness with respect to this predictor implies that the graduation prediction model does not depend on the likelihood that students are of a certain race.

- Harmful Stereotypes $\hat{p}_g^{(harm)}$: In the United States where this survey was conducted, there is a long history of school segregation. Today, a large fraction of Black and Latino students still attend schools that are predominantly non-white, and these schools provide fewer opportunities compared to other schools (Thompson Dorsey, 2013). We used a predictor $p_g$ based on the school's fraction of minority (non-white) students and the fraction of free lunch recipients.

- Prototypicality $\hat{p}_g^{(FCM)}$: We use a fuzzy clustering objective (Fuzzy c-means Bezdek et al. (1984)) to measure how typical relative to the cluster center for each race group according to cluster membership. We use PCA to reduce the feature dimension to 32 before identifying racial cluster centers for prediction. Fairness with respect to this stereotype predictor guarantees the college success prediction model does not depend on how typical each student is to their racial identity group.

Each of these predictors represents one particular operationalization of each of the three conceptualizations of stereotypes we presented in Section 4. The stereotypical associations predictor achieved 90.8% accuracy for race prediction, with a covariance $(\mathrm{Cov}(y, \hat{p}_g^{(all)}))$ of 0.018 and a significant correlation coefficient of 0.14 $(p < 10^{-7})$. In other words, this predictor is highly accurate at predicting race and demonstrates a significant correlation with the true label. The harmful stereotypes predictor, $\hat{p}_g^{(harm)}$, exhibits even stronger correlation with the target label while being a good proxy for race (e.g. accuracy: 91.1%, $\mathrm{Cov}(y, \hat{p}_g^{(harm)}) = 0.013$, and Pearson correlation $(\rho = 0.15, p = 1e - 8)$. This suggests that it might be difficult to achieve fairness notions such as demographic parity with respect to these stereotype predictors while preserving the overall accuracy.

**Fairness with respect to Stereotype Predictors**

Figure 5 illustrates that our post-processing Demographic Parity (DP) algorithm reduces demographic parity violations with respect to each stereotype predictor as well as to the combination of all three stereotype predictors. Unfortunately, there is a significant decrease in accuracy (from 83% to 63%) in predicting the probability of college graduation when enforcing demographic parity with respect to the harmful stereotype predictor $\hat{p}_g^{(harm)}$ as well as for the set of three predictors. For equal opportunity, we can achieve significant mitigation of $\mathrm{Cov}(p, \hat{p}_g | y = 1)$ without much impact on the overall accuracy for all stereotype predictors[15].

**Summary** In our case studies, we construct different stereotype predictors from various conceptual and operational choices. These predictors are associated with the outcome to different degrees: stereotype predictors with a close association to the outcome resulted in decreased performance for the overall model (DP $\hat{p}_g^{(harm)}$ for college success). Overall, we observed smaller impacts on model performance from equal opportunity interventions (even applying to all predictors together). These findings validate our theoretical results that the choice of stereotype predictor is important for preserving model performance.

## 8 Discussion

**Audit Predictors: A Unified View** Through the concept of *stereotype predictors*, our work proposes a unified view for measuring different notions of stereotype harms, and enforcing fairness with respect to these stereotypes. We observe that selecting different stereotype predictors to enforce fairness with respect to results in meaningfully different model behavior, impacting model performance to different degrees. In the two case studies we study, we show that non-trivial accuracy can be achieved even when several different stereotype predictors are jointly used to preserve fairness. Moreover, for equal opportunity notions where the true label is considered, there is little to no change in overall accuracy. Our approach is not exhaustive of all possible stereotype predictors but our contribution is to illustrate a setting where multiple stereotype predictors exist and can be considered jointly.

**Bridging Allocative and Representative Harms** Notions such as demographic parity and equal opportunity have predominately appeared in the literature of allocative harms (Hardt et al., 2016; Kleinberg et al., 2016). Our work examines the setting where representational harms such as stereotypes in training data may lead to downstream allocative harms when predictors are training on data containing stereotypes (Cheng et al., 2023a). As more language and image content are generated with foundation models, the representational issues in generated data may evolve into tangible biases in decision-making downstream. Here, the harmful beliefs are not held by humans exclusively but rather also held by models making decisions. Our work demonstrates different ways to prevent stereotypes in training data from leading to allocative harm.

---

[15]See Appendix B for full plots of results for this case study.

**Guidance for Stereotype Predictors**   In this work, we observed that enforcing fairness notions such as equal opportunity, equalized odds, and demographic parity with stereotype predictors that are highly correlated with the outcome can significantly reduce predictor accuracy. Similarly, applying these fairness notions with multiple distinct stereotype predictors can also negatively impact accuracy. In contrast, this limitation does not apply to multiaccuracy; a predictor can remain multiaccurate with respect to numerous stereotype predictors, even those closely tied to the outcome. Furthermore, for some stereotype predictors, different notions of fairness are inherently incompatible; our result is similar to but less sweeping than impossibility theorems for group-based fairness notions. As a result, when applying fairness with respect to stereotypes, it is important to select a single fairness definition and identify the best stereotype predictor for the stereotypes that we wish to protect.

## 9   Conclusion

Rather than prescribe a specific notion of stereotypes or associations from the plethora of existing definitions suggested by prior work, we focus our work in understanding the impacts of being fair with respect to these notions. To this end, we provide a unified view through introducing stereotype predictors, bridge notions of allocative and representational harms, and give guidance on the careful selection of stereotype predictors. Our goal is to provide an easy to understand analysis of stereotype predictors that will enable future works introducing new notions of stereotypes to apply our analysis.

**Limitations and Future Work**   While we do our best to give well-motivated examples for different notions of stereotypes, there are other concepts of stereotypes that we do not explicitly cover in our work. Furthermore, the normative decisions around which stereotypes to audit for and remove are not resolved by our work. These are decisions best left up to experts in each application domain. When the stereotype used as a stereotype predictor ($p_g$) is not carefully selected, imposing fairness requirements can cause harm rather than improvements. For example, we show that choosing a stereotype that is too correlated with the outcome can harm the accuracy. Specific stereotypes can also exclude some members of the minority group, the non-stereotypical ones according to the stereotype. A malicious choice of a stereotype predictor can result in further harm for subgroups and the overall population.

In the vast landscape of LLM-generated content, future work should explore the tradeoffs that arise in removing stereotype associations when more and more of the input data is synthetically generated or augmented. For example, we could study the implications of an LLM-enhanced set of resumes that may contain more or less stereotypical language on a downstream resume classifier.

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

## A Comparing and Contrasting Stereotype Predictors

There are many different conceptual and operational choices to make when creating a stereotype predictor $p_g$, each choice can produce a very different predictor. In Section 5, we show that enforcing fairness notions with respect to different stereotype predictors has different effects on the overall performance of the resulting models. Furthermore, the resulting models may treat individuals differently. Therefore, it is important to understand the implications of the different conceptual and operational choices. We highlight the consequences of some of these choices for the three example notions of stereotypes we described above:

- Stereotype association predictors are often more predictive of the group membership than predictors than the other two we suggest. When the features are sufficiently informative and the model is sufficiently expressive, $p_g$ might be approximately equal $g$. In this case, fairness with respect to $p_g$ is more akin to group notions of fairness with respect to $g$ than fairness with respect to stereotypes. However, stereotype association predictors can sometimes have undesirable properties; these predictors may use causal features with respect to the outcome (e.g. grades and college success). Using such a $p_g$ with respect to demographic parity to predict college success can lead to over or underpredicting the success of students with low grades.

- Building predictors for harmful stereotypes based on expert annotated features allows more judicious inclusion of which features we want decision-making models to be agnostic to. A socio-technical

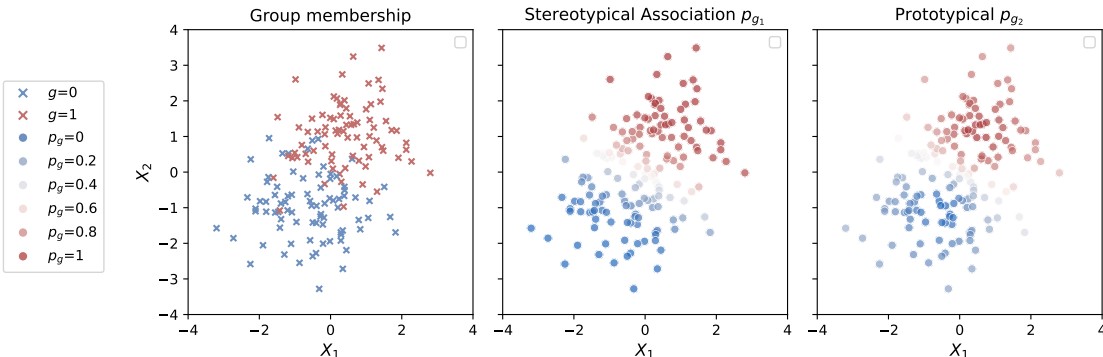

Figure 3: Synthetic Data: the group membership $g$ and two stereotype predictors $p_{g_1}, p_{g_2}$. The first, $p_{g_1}$ is using stereotypical association. It predicts the group membership better, and gives larger weight to $x_2$ comparing to $x_1$. The prototypical predictor, $p_{g_2}$, consider both features approximately equally. It give values closer to 0.5 to outliers, which might be undesired.

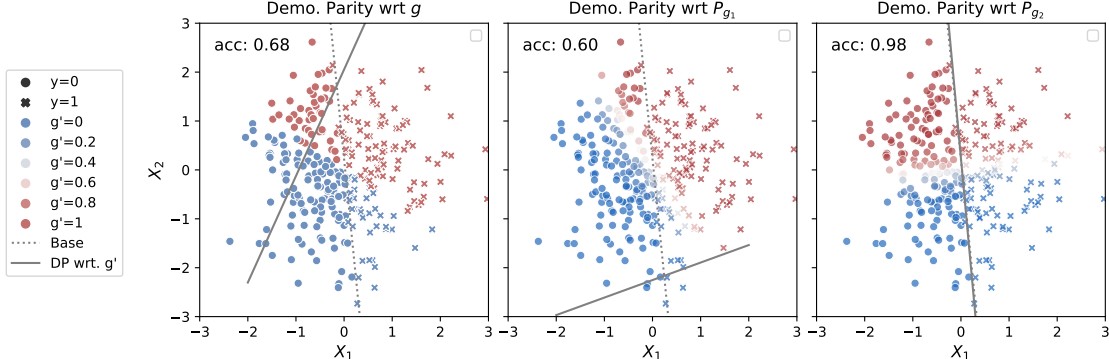

Figure 4: Synthetic Data: Demographic Parity (DP) wrt. group membership $g$ and their different stereotypes predictors $p_{g_1}$ and $p_{g_2}$ where $g' \in \{g, p_{g_1}, p_{g_2}\}$ for each plot. The decision boundary shifts significantly when achieving DP wrt. $p_{g_1}$ even more than $g$; resulting in a low accuracy predictor.

approach to identifying stereotypes is important when there is historical context to be considered. However, the necessity of expert identification of stereotypes makes auditing for these stereotypes challenging since expert advice may not be available or trustworthy (e.g. datasets may contain complex causal structures).

- Using a prototypical stereotype predictor $p_g$ by a clustering algorithm on a metric can encompass advantages and disadvantages from both of the two aforementioned predictors. Prototypical predictors can be constructed with a metric that is defined by experts who identify the more important features. This metric, and its weights, can then be applied across all features. When using prototypical predictor with a clustering algorithm, there is a risk of defining $p_g(x) \approx 0.5$, regardless of their features, see the synthetic data, Figure 3, for visual representation.

## A.1 Synthetic Data Experiments

### A.1.1 Stereotypical association and prototypical stereotype predictor

We give a visual demonstration of the difference between the types of stereotype predictors by using synthetic data. We generate a 2D data with two groups $g = 0, g = 1$. Both features depends on the group membership, but the dependency is much stronger in $x_2$. For $\mu = (0.5, 1)$ we sample $x \sim N(\mu, I)$ if $g = 1$ and $x \sim N(-\mu, I)$

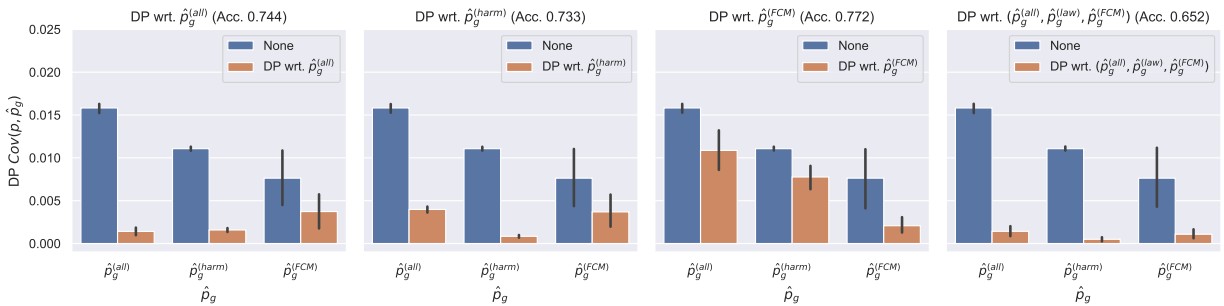

Figure 5: Demographic Parity Post-Processing with different race stereotype predictors for predicting degree attainment using the National Educational Longtitudinal Survey Dataset.

if $g = 0$. We run a logistic regression as a stereotypical association predictor, and fuzzy clustering algorithm with the Euclidean distance metric for the prototypical predictor..

For the stereotypical association predictor, we see on Figure 3 a clear separation in the prediction of the two groups. The predictor uses both features, but gives a larger weight to the more indicative feature, $x_2$. The prototypical predictor gives approximately the same weight to both features, as the weight is defined by the metric. It has the disadvantage of giving values rather close to 0.5 to outliers regardless of their direction.

### A.1.2 Demographic parity with respect to different stereotype predictors

We illustrate the effect of applying fairness notions to different stereotype predictors by using a 2D synthetic example for intuition. We generate data as follows: $x \sim N(0, I) \in \mathbb{R}^2$ and $y = \sigma(\beta x) + \epsilon$ where $\beta = [9, 1]$, $\epsilon \sim N(0, 0.1^2 I)$ and $\sigma$ is the sigmoid function. We then generate true group membership $g \in \{0, 1\}$ such that $\text{Cov}(y, g) \approx 0.11$. For stereotypes, we can consider the tuple $T(p_{g_i}) = \{(\text{Cov}(g, p_{g_i}), \text{Cov}(y, p_{g_i})\}$ where the first coordinate tells us how related the stereotype $g_i$ is to the true label $y$ and the second coordinate tells us how related the stereotype is $g_i$ is to the actual group membership $g$. We generate $p_{g_1} \in [0, 1]$ and $p_{g_2} \in [0, 1]$ such that $T(p_{g_1}) = \{0.38, 0.38\}$ and $T(p_{g_2}) = \{0.29, 0.00\}$. In other words, $p_{g_1}$ is associated with both the true label and the group membership while $p_{g_2}$ is only associated with group membership.

For demographic parity, Figure 4 illustrates that the choice of stereotype predictor $p_g$ can be indicative of the resulting accuracy. Achieving demographic parity with respect to $p_{g_1}$, a stereotype predictor with large covariance with both the group membership and label, comes at the cost of accuracy. However, since $p_{g_1}$ is closely related to $g$, achieving demographic parity with respect to $p_{g_1}$ also give improvements to demographic parity with respect to $g$. This effect also appears in the reverse direction.

## B  Case Study: Racial Stereotypes in College Success - Additional Plots

We include additional plots for the second case study in Figure 5 and Figure 6.

## C  Fairness Definitions

### C.1  Equivalence of Fairness Definitions

**Claim C.1.** *For a binary stereotype predictor $p_g : \mathcal{X} \to \{0, 1\}$:*

1. *Assuming $\Pr_{x \sim \mathcal{D}}[p_g(x) = 1]$ is bounded away from $\{0, 1\}$, $\alpha$-demographic parity with respect to stereotype in Definition 4.2 is equivalent to $\alpha / (\Pr_{x \sim \mathcal{D}}[p_g(x) = 1](1 - \Pr_{x \sim \mathcal{D}}[p_g(x) = 1]))$-demographic parity in Definition 3.1.*

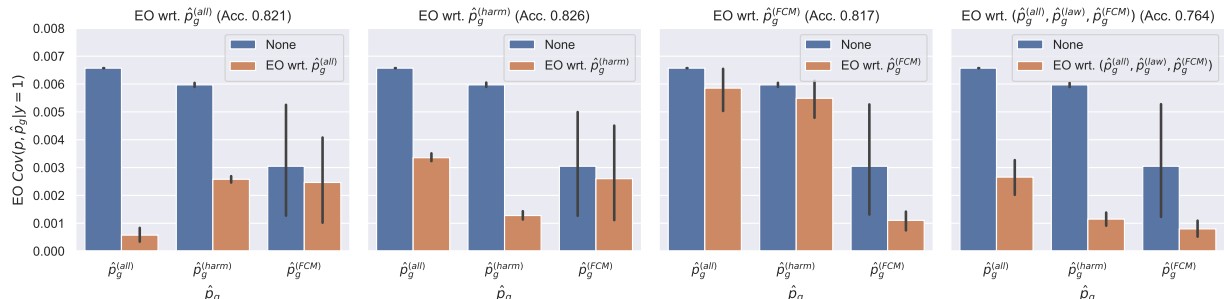

Figure 6: Equal Opportunity Post-Processing with different race stereotype predictors for predicting degree attainment using the National Educational Longitudinal Survey Dataset.

2. *Assuming* $\Pr_{x \sim \mathcal{D}}[p_g(x) = 1|y = 1]$ *is bounded away from* $\{0, 1\}$, $\alpha$-*equal opportunity with respect to stereotype in Definition 4.3 is equivalent to* $\alpha/(\Pr_{x \sim \mathcal{D}}[p_g(x) = 1|y = 1](1 - \Pr_{x \sim \mathcal{D}}[p_g(x) = 1|y = 1]))$-*equal opportunity in Definition 4.3.*

3. *An* $\alpha$-*multiaccurate predictor with respect to* $\{p_g, 1\}$ *satisfies* $2\alpha$ *covariance multiaccuracy with respect to* $p_g$. *A predictor* $p$ *satisfying* $\alpha$ *covariance multiaccuracy with respect to* $p_g$ *and* $|\mathbb{E}[p(x)] - \mathbb{E}[y]| \le \alpha$ *is* $2\alpha$-*multiaccurate with respect to* $p_g$.

*Proof.* Let $p_g : \mathcal{X} \to \{0, 1\}$ be a binary stereotype, let $p : \mathcal{X} \to [0, 1]$ be a predictor and $\alpha \in [0, 1]$ be some constant.

1. The group definition of $\gamma$-demographic parity, Definition 3.1, is

$$\left| \mathbb{E}_{x \sim \mathcal{D}}[p(x)|p_g(x) = 1] - \mathbb{E}_{x \sim \mathcal{D}}[p(x)|p_g(x) = 1] \right| \le \gamma.$$

Satisfying $\alpha$-demographic parity with respect to a stereotype, Definition 4.2, is $|\text{Cov}(p, p_g)| < \alpha$. We express the coraviance explicitly for a binary stereotype $p_g$:

$$\begin{aligned} |\text{Cov}(p, p_g)| &= \left| \mathbb{E}_{x \sim \mathcal{D}}[p(x)p_g(x)] - \mathbb{E}_{x \sim \mathcal{D}}[p(x)] \mathbb{E}_{x \sim \mathcal{D}}[p_g(x)] \right| \\ &= \left| \mathbb{E}_{x \sim \mathcal{D}}[p(x) \wedge p_g(x) = 1] - \mathbb{E}_{x \sim \mathcal{D}}[p(x)] \Pr[p_g(x) = 1] \right| \\ &= \left| \Pr_{x \sim \mathcal{D}}[p_g(x) = 1] \left( \mathbb{E}_{x \sim \mathcal{D}}[p(x)|p_g(x) = 1] - \mathbb{E}_{x \sim \mathcal{D}}[p(x)] \right) \right|. \end{aligned}$$

Denote $\Pr_{x \sim \mathcal{D}}[p_g(x) = 1] = \mu$, and we can write

$$\mathbb{E}_{x \sim \mathcal{D}}[p(x)] = \mu \mathbb{E}_{x \sim \mathcal{D}}[p(x)|p_g(x) = 1] + (1 - \mu) \mathbb{E}_{x \sim \mathcal{D}}[p(x)|p_g(x) \neq 1].$$

Substituting in the equation above, we get that for binary $p_g$, $|\text{Cov}(p, p_g)| \le \alpha$ is equivalent to $\mu(1 - \mu) |\mathbb{E}_{x \sim \mathcal{D}}[p(x)|p_g(x) = 1] - \mathbb{E}_{x \sim \mathcal{D}}[p(x)|p_g(x) \neq 1]| \le \alpha$, as required.

2. For equal opportunity, the proof is basically the same, only we condition on $y = 1$. The group definition of $\gamma$-equal opportunity, Definition 3.2, is

$$\left| \mathbb{E}_{x \sim \mathcal{D}}[p(x)|p_g(x) = 1, y = 1] - \mathbb{E}_{x \sim \mathcal{D}}[p(x)|p_g(x) = 1, y = 1] \right| \le \gamma.$$

For a stereotype predictor, we require that $|\text{Cov}(p, p_g|y = 1)| < \alpha$. Similar to the previous item, we write the covariance explicitly for a binary $p_g$. Let $\mu_1 = \Pr_{x \sim \mathcal{D}}[p_g(x) = 1|y = 1]$.

$$\begin{aligned} |\text{Cov}(p, p_g|y = 1)| &= \left| \mathbb{E}_{x \sim \mathcal{D}}[p(x)p_g(x)|y = 1] - \mathbb{E}_{x \sim \mathcal{D}}[p(x)|y = 1] \mathbb{E}_{x \sim \mathcal{D}}[p_g(x)|y = 1] \right| \\ &= \left| \mu_1 \left( \mathbb{E}_{x \sim \mathcal{D}}[p(x)|p_g(x) = 1, y = 1] - \mathbb{E}_{x \sim \mathcal{D}}[p(x)|y = 1] \right) \right|. \end{aligned}$$

Similarly, we express

$$\mathop{\mathbb{E}}_{x\sim\mathcal{D}}[p(x)|y=1] = \mu_1 \mathop{\mathbb{E}}_{x\sim\mathcal{D}}[p(x)|y=1, p_g(x)=1] + (1-\mu_1)\mathop{\mathbb{E}}_{x\sim\mathcal{D}}[p(x)|y=1, p_g(x)\neq 1]$$

and get that $|\mathrm{Cov}(p, p_g|y=1)|$ is equivalent to

$$\mu_1(1-\mu_1)\left|\mathop{\mathbb{E}}_{x\sim\mathcal{D}}[p(x)|y=1, p_g(x)=1] - \mathop{\mathbb{E}}_{x\sim\mathcal{D}}[p(x)|y=1, p_g(x)\neq 1]\right| \leq \alpha.$$

3. In Lemma 6.1, we prove that $\alpha$-multiaccuracy with respect to $\{p_g, 1\}$ implies $2\alpha$-covariance multiaccuracy (the proof appears on Appendix E.3). To finish the proof, we need to prove the other direction.

   Let $p$ be a predictor satisfying $\gamma$-covariance multiaccuracy, i.e. $|\mathrm{Cov}(p, p_g) - \mathrm{Cov}(y, p_g)| \leq \gamma$. We write the covariance explicitly:

$$\begin{aligned}
\gamma \geq &|\mathrm{Cov}(p, p_g) - \mathrm{Cov}(y, p_g)| \\
= &\left|\mathop{\mathbb{E}}_{x\sim\mathcal{D}}[p(x)p_g(x)] - \mathop{\mathbb{E}}_{x\sim\mathcal{D}}[p(x)]\mathop{\mathbb{E}}_{x\sim\mathcal{D}}[p_g(x)] - \mathop{\mathbb{E}}_{(x,y)\sim\mathcal{D}}[yp_g(x)] - \mathop{\mathbb{E}}_{(x,y)\sim\mathcal{D}}[y]\mathop{\mathbb{E}}_{x\sim\mathcal{D}}[p_g(x)]\right| \\
= &\left|\mathop{\mathbb{E}}_{(x,y)\sim\mathcal{D}}[p_g(x)(p(x)-y)] + \mathop{\mathbb{E}}_{x\sim\mathcal{D}}[p_g(x)](\mathop{\mathbb{E}}_{x\sim\mathcal{D}}[p(x)] - \mathop{\mathbb{E}}_{(x,y)\sim\mathcal{D}}[y])\right|
\end{aligned}$$

Since we assumed that $|\mathbb{E}_{x\sim\mathcal{D}}[p(x)] - \mathbb{E}_{(x,y)\sim\mathcal{D}}[y]| \leq \alpha$ we get that $\mathbb{E}_{(x,y)\sim\mathcal{D}}[p_g(x)(p(x)-y)] \leq 2\alpha$, as required. □

## C.2 Additional Definitions and Fairness Notions

**Definition C.2.** *A predictor $p: \mathcal{X} \to R$ with a discrete range is $\alpha$-calibrated, if for all $r \in R$ $|\mathbb{E}[y-r|p(x)=r]| \leq \alpha$.*

**Definition C.3.** *A predictor $p: \mathcal{X} \to R$ with a discrete range is $\alpha$-multicalibrated with respect to $p_g: \mathcal{X} \to [0,1]$, if for all $r \in R$, $|\mathbb{E}[p_g(x)(y-r)|p(x)=r]| \leq \alpha$.*

**Definition C.4.** *A predictor $p: \mathcal{X} \to [0,1]$ satisfies $\alpha$-equalized odds with respect to a group $g$ if*

   *1. $\left|\mathbb{E}_{(x,y)\sim\mathcal{D}}[p(x)|y=1, g(x)=1] - \mathbb{E}_{(x,y)\sim\mathcal{D}}[p(x)|y=1, g(x)=0]\right| \leq \alpha$.*

   *2. $\left|\mathbb{E}_{(x,y)\sim\mathcal{D}}[p(x)|y=0, g(x)=1] - \mathbb{E}_{(x,y)\sim\mathcal{D}}[p(x)|y=0, g(x)=0]\right| \leq \alpha$.*

Multicalibration is a stronger definition than multiaccuracy - every predictor $p$ that is multicalibrated with respect to a set $S$ is also multiaccurate with respect to it (up to change in the error parameter). Therefore, Lemma 5.6, showing that in some settings it is not possible to satisfy both demographic parity and multiaccuracy also holds for multicalibration. Similarly, equalized odds is a strengthening of equal opportunity. Therefore Lemma 5.2, showing that there exists cases that requiring equal opportunity can harm the accuracy, holds for equalized odds as well. The same is also true for Lemma 5.5, showing that in some cases, requiring equal opportunity with respect to multiple different stereotypes might force the predictor to be constant.

**Conditioning on Group Membership**

All fairness notions in Section 4.1 can also be defined when conditioning on $g=1$, as follows:

**Definition C.5.** *A predictor $p: \mathcal{X} \to [0,1]$ satisfies $\alpha$-demographic parity with respect to a stereotype predictor $p_g$ and group $g$ if:*

$$|\mathrm{Cov}(p, p_g|g=1| \leq \alpha.$$

**Definition C.6.** *A predictor $p: \mathcal{X} \to [0,1]$ satisfies $\alpha$-equal opportunity with respect to a stereotype predictor $p_g$ and group $g$ if:*

$$|\mathrm{Cov}(p, p_g|y=1, g=1)| \leq \alpha.$$

**Definition C.7.** *Let $\alpha \geq 0$, a predictor $p : \mathcal{X} \to [0,1]$ satisfies $\alpha$- covariance multiaccuracy with respect a set $S$ of stereotype predictors and a group $g$ if for every $p_g \in S$:*

$$|\text{Cov}(p, p_g | g = 1) - \text{Cov}(y, p_g | g = 1)| \leq \alpha.$$

When we use these definition, we apply the fairness requirement with respect to a stereotype $p_g$ only among the group members of $g$. For example, if we learn a predictor for sorting resume and require equal opportunity with respect to a stereotype predictor $p_g$, using Definition C.6 means we enforce it only between the women. Alternatively, it is possible to apply Definition C.6 twice - both for $x \in g$ and for $x \notin g$.

In the paper we chose to use the definitions on Section 4.1 and not to condition on $g = 1$, but the results in our paper can be easily extended to Definitions C.5,C.6 and C.7. In order to extended our results to these definitions, we need to rewrite the claims and proofs when conditioning on $g = 1$. In Appendix D.2 we give an example of a distribution and a few stereotype predictors, such that requiring demographic parity or equal opportunity with respect to all of them causes the predictor to be constant. In this section we also describe shortly how it can be extended to an example also when conditioning over $g = 1$.

# D Proof for Choosing the Stereotype Predictor

## D.1 Choosing a Single Stereotype Predictor

In this section we prove Claims 5.1 and 5.2.

*Proof of Lemma 5.1.* Let $p_g$ be a predictor as described. Let $p^* : \mathcal{X} \to [0,1]$ be the predictor of the underlying distribution of $y$ given $x$. Then $\text{Cov}(p_g, p^*) > \gamma$. Let $p$ be any predictor satisfying $\text{Cov}(p_g, p) \leq \alpha$. From the claim statement, $\text{Cov}(p_g, p^*) - \text{Cov}(p_g, p) > \gamma - \alpha$. Using the linearity of expectation, $\text{Cov}(p_g, p^* - p) > \gamma - \alpha$. This gives a lower bound on the $\ell_2$ norm of $p^* - p$:

$$\text{Cov}(p_g, p^* - p) \leq \sqrt{\text{Var}(p_g)\text{Var}(p^* - p)}$$

$$\text{Var}(p^* - p) \geq \frac{(\gamma - \alpha)^2}{\text{Var}(p_g)} \geq 4(\gamma - \alpha)^2.$$

The factor 4 is because the variance of a bounded random variables in $[0,1]$ is at most $1/4$. This means that $\mathbb{E}_{x \sim \mathcal{D}}[(p^*(x) - p(x))^2] \geq 4(\gamma - \alpha)^2$. The $\ell_2$ loss is proper, therefore given that $y \sim \text{Ber}(p^*(x))$, the predictor $p^*$ itself achieves the lowest possible loss. For every $x$ we have that the expected $\ell_2$-loss given that $x$ is chosen is

$$
\begin{aligned}
\mathbb{E}_{y \sim \text{Ber}(p^*(x))}[(y - p(x))^2] &= \mathbb{E}_{y \sim \text{Ber}(p^*(x))}[(y - p^*(x) + p^*(x) - p(x))^2] \\
&= \mathbb{E}_{y \sim \text{Ber}(p^*(x))}[(y - p^*(x))^2] + 2 \mathbb{E}_{y \sim \text{Ber}(p^*(x)}[(y - p^*(x))(p^*(x) - p(x))] \\
&\quad + \mathbb{E}_{y \sim \text{Ber}(p^*(x)}[(p^*(x) - p(x))^2] \\
&= \mathbb{E}_{y \sim \text{Ber}(p^*(x))}[(y - p^*(x))^2] + (p^*(x) - p(x))^2.
\end{aligned}
\tag{1}
$$

In the last equality, we use the fact that the distribution is only on $y$, and therefore $(p^*(x) - p(x))$ is constant, and $\mathbb{E}_y[y - p^*(x)] = 0$. The above argument is a proof that the $\ell_2$ loss is proper.

Applying the above equality on $x \sim \mathcal{D}$ we have that $\mathbb{E}_{(x,y) \sim \mathcal{D}}[(y - p(x))^2] \geq \mathbb{E}_{(x,y) \sim \mathcal{D}}[(y - p^*(x))^2] + \mathbb{E}_{(x,y) \sim \mathcal{D}}[(p^*(x) - p(x))^2]$, and by using $\mathbb{E}_{x \sim \mathcal{D}}[(p^*(x) - p(x))^2] \geq 4(\gamma - \alpha)^2$ we finish the proof. □

*Proof of Lemma 5.2.* From the conditions of the lemma, we have that $\text{Cov}(p^*, p_g | y = 1) > \gamma$ and $\text{Cov}(p, p_g | y = 1) < \alpha$. We apply Claim D.1 with $\beta = \gamma - \alpha$, and get that $\mathbb{E}_{x \sim \mathcal{D}}[|p^*(x) - p(x)|] \geq (\gamma - \alpha) \mathbb{E}_{(x,y) \sim \mathcal{D}}[y]$. We use this to bound the loss:

$$\mathbb{E}_{x \sim \mathcal{D}}[(p_1(x) - p_2(x))^2] = \mathbb{E}_{x \sim \mathcal{D}}[(f(x))^2] \geq \left(\mathbb{E}_{x \sim \mathcal{D}}[|f(x)|]\right)^2 \geq (\gamma - \alpha)^2 \left(\mathbb{E}_{x \sim \mathcal{D}}[y]\right)^2.$$

Using the same argument as in equation (1), we have that

$$\mathop{\mathbb{E}}_{(x,y)\sim\mathcal{D}}[(y-p(x))^2] \leq \mathop{\mathbb{E}}_{(x,y)\sim\mathcal{D}}[(p^*(x)-y)^2] + \mathop{\mathbb{E}}_{(x,y)\sim\mathcal{D}}[(p^*(x)-p(x))^2]$$

$$\leq \mathop{\mathbb{E}}_{(x,y)\sim\mathcal{D}}[(p^*(x)-y)^2] + (\gamma-\alpha)^2 \left(\mathop{\mathbb{E}}_{x\sim\mathcal{D}}[y]\right)^2.$$

$\square$

*Proof of Lemma 5.3.* From the conditions of the lemma, we have that $\mathrm{Cov}(p_{opt}, p_g | y = 1) > \gamma$ and $\mathrm{Cov}(p', p_g | y = 1) < \alpha$.

The predictor $p_{opt}$ is chosen to be the predictor minimizing the $\ell_2$ loss, assuming that $y$ is distributed according to $\mathrm{Ber}(p(x))$. Since $p'$ is a post-processing of $p, p_g$ we have that

$$\mathop{\mathbb{E}}_{(x,y)\sim\mathcal{D}}[(y-p_{opt}(x))^2] \leq \mathop{\mathbb{E}}_{(x,y)\sim\mathcal{D}}[(y-p'(x))^2] \tag{2}$$

We lower bound the loss. Using the triangle inequality,

$$\mathop{\mathbb{E}}_{(x,y)\sim\mathcal{D}}[(y-p'(x))^2] \geq \mathop{\mathbb{E}}_{(x,y)\sim\mathcal{D}}[(p(x)-p_{opt}(x))^2] - \mathop{\mathbb{E}}_{(x,y)\sim\mathcal{D}}[(y-p_{opt}(x))^2]. \tag{3}$$

Applying Claim D.1, we have that $\mathbb{E}_{(x,y)\sim\mathcal{D}}[(p(x)-p_{opt}(x))^2] \geq (\gamma-\alpha)^2 \left(\mathbb{E}_{(x,y)\sim\mathcal{D}}[y]\right)^2$. Therefore, we get that

$$\mathop{\mathbb{E}}_{(x,y)\sim\mathcal{D}}[(y-p'(x))^2] \geq \frac{(\gamma-\alpha)^2}{2} \left(\mathop{\mathbb{E}}_{(x,y)\sim\mathcal{D}}[y]\right)^2. \tag{4}$$

$\square$

**Claim D.1.** *Let $p_1, p_2, p_g : \mathcal{X} \to [0,1]$ be three predictors, and assume that*

$$|\mathrm{Cov}(p_1, p_g | y = 1) - \mathrm{Cov}(p_1, p_g | y = 1)| > \beta,$$

*Then we have that $\mathbb{E}_{x\sim\mathcal{D}}[|p_1(x) - p_2(x)|] \geq \beta \, \mathbb{E}_{x\sim\mathcal{D}}[y]$.*

*Proof.* Let $p_1, p_2, p_g$ be as in the claim, and let $f(x) = p_1(x) - p_2(x)$. Since covariance is additive, we have that $|\mathrm{Cov}(f, p_g | y = 1)| > \beta$. Using Claim D.2, we write $\mathrm{Cov}(f, p_g | y = 1)$ explicitly.

$$\mathrm{Cov}(f, p_g | y = 1) = \frac{\mathbb{E}_{x,y\sim\mathcal{D}}[p^*(x)f(x)p_g(x)]}{\mathbb{E}_{x,y\sim\mathcal{D}}[y]} - \frac{\mathbb{E}_{x,y\sim\mathcal{D}}[p^*(x)f(x)]}{\mathbb{E}_{x,y\sim\mathcal{D}}[y]} \frac{\mathbb{E}_{x,y\sim\mathcal{D}}[p^*(x)p_g(x)]}{\mathbb{E}_{x,y\sim\mathcal{D}}[y]}$$

Both $p^*$ and $p_g$ are bounded in $[0,1]$, we use this fact to upper bound the covariances:

$$|\mathrm{Cov}(f, p_g | y = 1)| = \frac{1}{\mathbb{E}_{x,y\sim\mathcal{D}}[y]} \left| \mathop{\mathbb{E}}_{x,y\sim\mathcal{D}} \left[ p^*(x)f(x) \left( p_g(x) - \frac{\mathbb{E}_{x,y\sim\mathcal{D}}[p^*(x)p_g(x)]}{\mathbb{E}_{x,y\sim\mathcal{D}}[y]} \right) \right] \right|$$

$$\leq \frac{1}{\mathbb{E}_{x,y\sim\mathcal{D}}[y]} \mathop{\mathbb{E}}_{x,y\sim\mathcal{D}} \left[ |p^*(x)f(x)| \left| p_g(x) - \frac{\mathbb{E}_{x,y\sim\mathcal{D}}[p^*(x)p_g(x)]}{\mathbb{E}_{x,y\sim\mathcal{D}}[y]} \right| \right]$$

$$\leq \frac{\mathbb{E}_{x,y\sim\mathcal{D}}[|f(x)|]}{\mathbb{E}_{x,y\sim\mathcal{D}}[y]},$$

as required. $\square$

**Claim D.2.** *Let $(x, y) \sim \mathcal{D}$ such that $\mathbb{E}_{x\sim\mathcal{D}}[y] > 0$ and denote by $p^* : \mathcal{X} \to [0,1]$ the conditional distribution of $y$ given $x$. Let $f : \mathcal{X} \to [0,1]$ be any function. Then*

$$\mathop{\mathbb{E}}_{x\sim\mathcal{D}}[f(x) | y = 1] = \frac{\mathbb{E}_{x\sim\mathcal{D}}[f(x)p^*(x)]}{\mathbb{E}[y]}.$$

*Proof.* Writing the expectation explicitly:

$$
\mathbb{E}_{x \sim \mathcal{D}}[f(x)|y = 1] = \sum_{x' \in \mathcal{X}} \Pr_{x \sim \mathcal{D}}[x = x'|y = 1]f(x')
$$

$$
= \sum_{x' \in \mathcal{X}, p^*(x)} \frac{\Pr_{x \sim \mathcal{D}}[x = x', y = 1]}{\Pr_{(x,y) \sim \mathcal{D}}[y = 1]} f(x')
$$

$$
= \sum_{x' \in \mathcal{X}} \frac{Pr_{x \sim \mathcal{D}}[x = x']p^*(x')}{\Pr_{(x,y) \sim \mathcal{D}}[y = 1]} f(x')
$$

$$
= \frac{1}{\mathbb{E}_{(x,y) \sim \mathcal{D}}[y]} \sum_{x' \in \mathcal{X}} \Pr_{x \sim \mathcal{D}}[x = x']p^*(x)f(x)
$$

$$
= \frac{\mathbb{E}_{x \sim \mathcal{D}}[p^*(x)f(x)]}{\mathbb{E}_{(x,y) \sim \mathcal{D}}[y]}.
$$

$\square$

### D.2 Choosing Multiple Predictors Bounds

We give an example of a simple distribution and stereotype predictors, such that if we require $p$ to satisfy demographic parity or equal opportunity with respect to all stereotype predictors $p_g$'s, then $p$ would have to be constant. We remark that all stereotype predictors $p_g$ are calibrated, i.e. $\mathbb{E}_{x \sim \mathcal{D}}[g(x)|p_{g_i}(x) = r] = r$. We start by describing the example, and then prove that it satisfies the properties.

**Example D.3.** *Let $\mathcal{X}$ the population be divided into $4$ different equal sized groups, deboted by $S_1, \ldots, S_4$. Suppose that $y$ and $g$ are defined by $p^*$ and $p_g^*$ respectively. Let $p_g^1, \ldots, p_g^4$ be four different stereotype predictors for $g$, defined as follows.*

| Group | $p^*$ | $p_g^*$ | $p_{g_1}$ | $p_{g_2}$ | $p_{g_3}$ | $p_{g_4}$ |
|-------|-------|---------|-----------|-----------|-----------|-----------|
| $S_1$ | 0.8 | 0.8 | 0.8 | 0.4 | 0.6 | 0.6 |
| $S_2$ | 0.5 | 0.8 | 0.4 | 0.8 | 0.6 | 0.6 |
| $S_3$ | 0.5 | 0.2 | 0.4 | 0.4 | 0.2 | 0.6 |
| $S_4$ | 0.2 | 0.2 | 0.4 | 0.4 | 0.6 | 0.2 |

(5)

**Claim D.4.** *Any predictor $p : \mathcal{X} \to [0, 1]$ that satisfies $\alpha$-demographic parity with respect to $p_{g_1}, \ldots, p_{g_4}$ must satisfy $\mathbb{E}_{x \sim \mathcal{D}}[|p(x) - \mu|] \leq 10\alpha$, where $\mu = \mathbb{E}_{x \sim \mathcal{D}}[p(x)]$.*

That is, any predictor $p$ that satisfies demographic parity with respect to all stereotypes is rather close to the constant predictor.

*Proof.* Let $p : \mathcal{X} \to [0, 1]$ a predictor satisfying $\text{Cov}(p_{g_i}, p) = 0$ for $i \in [4]$. We note that since there are only 4 groups, the predictor $p$ can be described by 4 numbers which are the value of $p$ on the parts. Denote by $\eta_i = p(x)$ for $x \in S_i$. We start from assuming $\alpha = 0$, for simplicity.

$$
0 = \text{Cov}(p_{g_1}, p) = \frac{1}{4}(0.8\eta_1 + 0.4(\eta_2 + \eta_3 + \eta_4)) - \frac{1}{4}(\eta_1 + \eta_2 + \eta_3 + \eta_4)0.5.
$$

$$
0 = \text{Cov}(p_{g_2}, p) = \frac{1}{4}(0.8\eta_2 + 0.4(\eta_1 + \eta_3 + \eta_4)) - \frac{1}{4}(\eta_1 + \eta_2 + \eta_3 + \eta_4)0.5.
$$

$$
0 = \text{Cov}(p_{g_3}, p) = \frac{1}{4}(0.2\eta_3 + 0.6(\eta_1 + \eta_2 + \eta_4)) - \frac{1}{4}(\eta_1 + \eta_2 + \eta_3 + \eta_4)0.5.
$$

$$
0 = \text{Cov}(p_{g_4}, p) = \frac{1}{4}(0.2\eta_4 + 0.4(\eta_1 + \eta_2 + \eta_3)) - \frac{1}{4}(\eta_1 + \eta_2 + \eta_3 + \eta_4)0.5.
$$

Solving these four equations we get that for all $i \in [4]$, $\eta_i = \frac{1}{3}\sum_{i' \neq i} \eta_{i'}$, and the only solution is $\eta_1 = \eta_2 = \eta_3 = \eta_4$.

Assuming we are required only for $\alpha$-demographic parity, for some small constant $\alpha > 0$, then all of the above qualities are in fact inequalities, i.e. we have that for all $i \in [4]$,

$$\left| 3\eta_i - \sum_{i' \neq i} \eta_{i'} \right| \leq 40\alpha.$$

Let $\mu$ be the expected value of $p$, and assume that $p$ is $\epsilon$-far from being constant in $\ell_1$ distance. Let $i$ be the indicator to the set $S_i$ on which $|\eta_i - \mu|$ is maximal. Then we have that $|\eta_i - \mu| \geq \epsilon$. This implies that

$$\left| 3\eta_i - \sum_{i' \neq i} \eta_{i'} \right| = |3\eta_i - (4\mu - \eta_i)| = |4\eta_i - 4\mu| \geq 4\epsilon.$$

Therefore, to satisfy the demographic parity requirements, we must have that

$$4\epsilon \leq \left| 3\eta_i - \sum_{i' \neq i} \eta_{i'} \right| \leq 40\alpha.$$

To satisfy the $\alpha$-demographic parity requirement, we need $\epsilon < 10\alpha$, i.e. $\mathbb{E}_{x \sim \mathcal{D}}[\ell_1(p, \mu)] \leq 10\alpha$. $\qquad\square$

We show a similar claim when requiring equal opportunity.

**Claim D.5.** *Any predictor $p : \mathcal{X} \to [0, 1]$ that satisfies $\alpha$-equal opportunity with respect to $p_{g_1}, \ldots, p_{g_4}$ must satisfy $\mathbb{E}_{x \sim \mathcal{D}}[|p(x) - \mu|] \leq 100\alpha$, where $\mu = \mathbb{E}_{x \sim \mathcal{D}}[p(x)]$.*

*Proof.* Let $p$ be a predictor that satisfies $\alpha$-equal opportunity with respect to $p_{g_1}, \ldots, p_{g_4}$. That is, for $i \in [4]$ we have $\mathrm{Cov}(p, p_g| = 1) \leq \alpha$. Similarly to the previous proof, we denote by $\eta_1, \ldots, \eta_4$ the values of $p$ on parts $S_1, \ldots, S_4$. Conditioning on $y = 1$ changes the distribution over the parts $S_1, \ldots, S_4$ to:

$$\beta_1 = \Pr[x \in S_1 | y = 1] = 0.4, \qquad\qquad \beta_2 = \Pr[x \in S_2 | y = 1] = 0.25,$$
$$\beta_3 = \Pr[x \in S_3 | y = 1] = 0.25, \qquad\qquad \beta_4 = \Pr[x \in S_4 | y = 1] = 0.1. \qquad (6)$$

Using these we can calculate the conditional expectation and receive that $\mathrm{Cov}(p_{g_i}, p | y = 1) = \sum_i \beta_i \eta_i p_{g_i}(S_i) - (\sum_i \beta_i \eta_i)(\sum_i \beta_i p_{g_i}(S_i))$. Therefore, we have that $\mathrm{Cov}(p, p_{g_i} | y = 1) = \sum_i \beta_i \eta_i (p_{g_i}(S_i) - \mathbb{E}[p_{g_i} | y = 1])$. By substituting the values of $\beta_i$ and the expectation we have that:

$$\mathrm{Cov}(p, p_{g_1} | y = 1) = 0.096\eta_1 - 0.04\eta_2 - 0.04\eta_3 - 0.016\eta_4$$
$$\mathrm{Cov}(p, p_{g_2} | y = 1) = -0.04\eta_1 + 0.075\eta_2 - 0.025\eta_3 - 0.01\eta_4$$
$$\mathrm{Cov}(p, p_{g_3} | y = 1) = 0.04\eta_1 + 0.025\eta_2 - 0.075\eta_3 + 0.01\eta_4$$
$$\mathrm{Cov}(p, p_{g_4} | y = 1) = 0.016\eta_1 + 0.01\eta_2 + 0.01\eta_3 - 0.036\eta_4.$$

If we require the covariance to be 0, then again the only solution is the constant predictor $\eta_i = \mu$ for all $i$.

If we only require $\alpha$-equal opportunity, then from the covariance with $p_{g_1}, p_{g_2}$ we have that $|\eta_2 - \eta_3| \leq 20\alpha$ and from other two $|\eta_1 - \eta_4| \leq 32\alpha$. Using it on the first equation we have that $|\eta_1 - \eta_2| \leq 100\alpha$. Therefore, together we have that $\alpha$-equal opportunity also implies that the predictor is rather close to constant. $\qquad\square$

We remark that the counter example also works when requiring equal-opportunity when conditionong on $y = 1$ and $g = 1$. In this case, we have a similar expression for the covariance, only with different $\beta_i$'s:

$$\beta_1 = \Pr[x \in S_1 | y = 1, g = 1] = 0.54, \qquad\qquad \beta_2 = \Pr[x \in S_2 | y = 1, g = 1] = 0.34,$$
$$\beta_3 = \Pr[x \in S_3 | y = 1, g = 1] = 0.088, \qquad\qquad \beta_4 = \Pr[x \in S_4 | y = 1, g = 1] = 0.034. \qquad (7)$$

And repeating a similar argument also results in 4 different linear equations where the constant predictor is the only solution.

### D.3 Using Different Fairness Definitions

In this section we use Example D.3 to show that when enforcing different fairness definitions, it is possible to force the predictor to be constant, or that there will be no solution satisfying all requirements.

**Claim D.6.** *For $\alpha < 0.0075$, no predictor $p : \mathcal{X} \to [0,1]$ can satisfy $\alpha$-multiaccuracy with respect to $\{p_{g_1}, p_{g_2}\}$ and $\alpha$-demographic parity with respect to $p_{g_3}, p_{g_4}$.*

*Proof.* Let $p : \mathcal{X} \to [0,1]$ satisfies the conditions in the claims, and denote by $\eta_1, \dots, \eta_4$ be the predictor on each of the sets $S_1, \dots, S_4$. We write the multiaccuracy and demographic parity requirements explicitly: $|\mathrm{Cov}(p_{g_i}, p)| = \frac{1}{40}\left(3\eta_i - \sum_{i' \neq i} \eta_i\right)$. We also have that $\mathrm{Cov}(p_{g_1}, y) = 0.03, \mathrm{Cov}(p_{g_2}, y) = \mathrm{Cov}(p_{g_3}, y) = 0$. For $p_{g_4}$ we require demographic parity, i.e. that $|\mathrm{Cov}(p_{g_4}, p)| < \alpha$.

Denote by $\mu = \mathbb{E}_{x \sim \mathcal{D}}[p(x)] = \frac{1}{4}\sum_{i=1}^{4} \eta_i$, then we have that:

$$
\begin{aligned}
|\eta_1 - \mu| &\leq 0.3 + 10\alpha, \\
|\eta_2 - \mu| &\leq 10\alpha, \\
|\eta_3 - \mu| &\leq 10\alpha, \\
|\eta_4 - \mu| &\leq 10\alpha.
\end{aligned}
$$

If $40\alpha < 0.3$, then no values of $\eta_1, \eta_2, \eta_3, \eta_4$ can satisfy all requirements. $\square$

**Claim D.7.** *Any predictor $p : \mathcal{X} \to [0,1]$ that satisfies $\alpha$-equal opportunity with respect to $p_{g_1}, p_{g_2}$ and demographic parity with respect to $p_{g_3}, p_{g_4}$ must satisfy $\mathbb{E}_{x \sim \mathcal{D}}[|p(x) - \mu|] \leq 40\alpha$, where $\mu = \mathbb{E}_{x \sim \mathcal{D}}[p(x)]$.*

*Proof.* The proof is done by using the conditions of the demographic parity and equal opportunity: Similarly to before, let $p : \mathcal{X} \to [0,1]$ be a predictor satisfying the claim's conditions. Let $\eta_1, \dots, \eta_4$ the value of the predictor on sets $S_1, \dots, S_4$. Let $\mu = \frac{1}{4}\sum_{i=1}^{4} \eta_i$ be the expected value of $p$. Then it should satisfy:

$$
\begin{aligned}
|\eta_1 - \mu| &\leq 10\alpha, \\
|\eta_2 - \mu| &\leq 10\alpha, \\
|4\eta_1 + 2.5\eta_- 7.5\eta_3 + \eta_4| &\leq 100\alpha, \\
|1.6\eta_1 + \eta_2 + \eta_3 - 3.6\eta_4| &\leq 100\alpha.
\end{aligned}
$$

By solving all of the equations, we find that the only solution is when all $\eta_i$ are no more than $40\alpha$ far from the expectation. $\square$

### D.4 Calibration and Equalized Odds

*Proof.* The main idea in the proof is that we express $\mathrm{Cov}(y, p_g)$ as a weighted sum of $\mathrm{Cov}(p, p_g|y = 1)$ and $\mathrm{Cov}(p, p_g|y = 0)$, with an additional additive term. Then, we show that it is not possible that $\mathrm{Cov}(y, p_g)$ is large and both $\mathrm{Cov}(p, p_g|y = 1)$ and $\mathrm{Cov}(p, p_g|y = 0)$ are small.

Let $\mu_1 = \mathrm{Pr}_{(x,y) \sim \mathcal{D}}[y = 1] = \mathbb{E}_{(x,y) \sim \mathcal{D}}[y]$. In order to express $\mathrm{Cov}(y, p_g)$ using $\mathrm{Cov}(p, p_g|y = 1)$ and $\mathrm{Cov}(p, p_g|y = 0)$, we calculate the following terms:

$$
\mathbb{E}_{(x,y) \sim \mathcal{D}}[p_g(x)p(x)] = \mu_1 \mathbb{E}_{(x,y) \sim \mathcal{D}}[p_g(x)p(x)|y = 1] + (1 - \mu_1) \mathbb{E}_{(x,y) \sim \mathcal{D}}[p_g(x)p(x)|y = 0]. \tag{8}
$$

We do the same for the second term in the covariance:

$$
\mathop{\mathbb{E}}_{(x,y)\sim\mathcal{D}}[p_g(x)] \mathop{\mathbb{E}}_{(x,y)\sim\mathcal{D}}[p(x)] = \left(\mu_1 \mathop{\mathbb{E}}_{(x,y)\sim\mathcal{D}}[p_g(x)|y=1] + (1-\mu_1)\mathop{\mathbb{E}}_{(x,y)\sim\mathcal{D}}[p_g(x)|y=0]\right)
$$
$$
\cdot\left(\mu_1 \mathop{\mathbb{E}}_{(x,y)\sim\mathcal{D}}[p(x)|y=1] + (1-\mu_1)\mathop{\mathbb{E}}_{(x,y)\sim\mathcal{D}}[p(x)|y=0]\right)
$$
$$
=\mu_1^2 \mathop{\mathbb{E}}_{(x,y,g)\sim\mathcal{D}}[p_g(x)|y=1]\mathop{\mathbb{E}}_{(x,y)\sim\mathcal{D}}[p(x)|y=1]
$$
$$
+ (1-\mu_1)^2 \mathop{\mathbb{E}}_{(x,y)\sim\mathcal{D}}[p_g(x)|y=0]\mathop{\mathbb{E}}_{(x,y)\sim\mathcal{D}}[p(x)|y=0]
$$
$$
+ \mu_1(1-\mu_1)\mathop{\mathbb{E}}_{(x,y)\sim\mathcal{D}}[p_g(x)|y=0]\mathop{\mathbb{E}}_{(x,y)\sim\mathcal{D}}[p(x)|y=1]
$$
$$
+ \mu_1(1-\mu_1)\mathop{\mathbb{E}}_{(x,y)\sim\mathcal{D}}[p_g(x)|y=1]\mathop{\mathbb{E}}_{(x,y)\sim\mathcal{D}}[p(x)|y=0].
$$

Putting it all together, we have that

$$
\mathrm{Cov}(p,p_g) = \mu_1\mathrm{Cov}(p,p_g|y=1) + (1-\mu_1)\mathrm{Cov}(p,p_g|y=0)
$$
$$
+ \mu_1(1-\mu_1)\sum_{b,b'\in\{0,1\}}(-1)^{b+b'}\mathop{\mathbb{E}}_{(x,y)\sim\mathcal{D}}[p_g(x)|y=b]\mathop{\mathbb{E}}_{(x,y)\sim\mathcal{D}}[p(x)|y=b'] \tag{9}
$$

We want to bound the last sum. Using Claim D.2, we have:

$$
\mathop{\mathbb{E}}_{(x,y)\sim\mathcal{D}}[p(x)|y=1] = \frac{\mathbb{E}_{(x,y)\sim\mathcal{D}}[p^*(x)p(x)]}{\mu_1}
$$
$$
\mathop{\mathbb{E}}_{(x,y)\sim\mathcal{D}}[p_g(x)|y=1] = \frac{\mathbb{E}_{(x,y)\sim\mathcal{D}}[p^*(x)p_g(x)]}{\mu_1}
$$
$$
\mathop{\mathbb{E}}_{(x,y)\sim\mathcal{D}}[p(x)|y=0] = \frac{\mathbb{E}_{(x,y)\sim\mathcal{D}}[(1-p^*(x))p(x)]}{1-\mu_1}
$$
$$
\mathop{\mathbb{E}}_{(x,y)\sim\mathcal{D}}[p_g(x)|y=1] = \frac{\mathbb{E}_{(x,y)\sim\mathcal{D}}[(1-p^*(x))p_g(x)]}{1-\mu_1}
$$

Therefore, the sum above corresponds to

$$
\mu_1(1-\mu_1)\sum_{b,b'\in\{0,1\}}(-1)^{b+b'}\mathop{\mathbb{E}}_{(x,y)\sim\mathcal{D}}[p_g(x)|y=b]\mathop{\mathbb{E}}_{(x,y)\sim\mathcal{D}}[p(x)|y=b'] =
$$
$$
\frac{1-\mu_1}{\mu_1}\mathop{\mathbb{E}}_{(x,y)\sim\mathcal{D}}[p^*(x)p(x)]\mathop{\mathbb{E}}_{(x,y)\sim\mathcal{D}}[p^*(x)p_g(x)] - \mathop{\mathbb{E}}_{(x,y)\sim\mathcal{D}}[p^*(x)p(x)]\mathop{\mathbb{E}}_{(x,y)\sim\mathcal{D}}[(1-p^*(x))p_g(x)]
$$
$$
- \mathop{\mathbb{E}}_{(x,y)\sim\mathcal{D}}[(1-p^*(x))p(x)]\mathop{\mathbb{E}}_{(x,y)\sim\mathcal{D}}[p(x)^*p_g(x)] +
$$
$$
+ \frac{\mu_1}{1-\mu_1}\mathop{\mathbb{E}}_{(x,y)\sim\mathcal{D}}[(1-p^*(x))p(x)]\mathop{\mathbb{E}}_{(x,y)\sim\mathcal{D}}[(1-p^*(x))p_g(x)]
$$
$$
= \frac{\mathbb{E}_{(x,y)\sim\mathcal{D}}[p^*(x)p_g(x)]}{\mu_1(1-\mu_1)}\left(\mathop{\mathbb{E}}_{(x,y)\sim\mathcal{D}}[p^*(x)p(x)] - \mu_1\mathop{\mathbb{E}}_{(x,y)\sim\mathcal{D}}[p^*(x)]\right)
$$
$$
+ \frac{\mathbb{E}_{(x,y)\sim\mathcal{D}}[p_g(x)]}{1-\mu_1}\left(-\mathop{\mathbb{E}}_{(x,y)\sim\mathcal{D}}[p^*(x)p(x)] + \mu_1\mathop{\mathbb{E}}_{(x,y)\sim\mathcal{D}}[p(x)]\right)
$$
$$
= \frac{\mathrm{Cov}(p^*,p_g)\mathrm{Cov}(p^*,p)}{\mu_1(1-\mu_1)}.
$$

Our assumption is that $y$ is chosen as a Bernuli random variable from a distribution $p^* : \mathcal{X} \to [0,1]$. Therefore, given $p^*(x)$ the random variable $y$ is independent in $p(x), p_g(x)$, and $\mathrm{Cov}(p^*,p_g) = \mathrm{Cov}(y,p_g)$, $\mathrm{Cov}(p^*,p) = \mathrm{Cov}(y,p)$.

Putting it all together, we have that

$$\mathrm{Cov}(p, p_g) = \mu_1 \mathrm{Cov}(p, p_g | y = 1) + (1 - \mu_1) \mathrm{Cov}(p, p_g | y = 0) + \frac{\mathrm{Cov}(y, p_g) \mathrm{Cov}(y, p)}{\mu_1 (1 - \mu_1)}. \tag{10}$$

From multiaccuracy and the fact that $\mathbb{E}[p(x)]$ is similar to $\mathbb{E}[y]$,

$$\begin{aligned}
|\mathrm{Cov}(y, p_g) - \mathrm{Cov}(p, p_g)| &= \mathrm{Cov}(y - p, p_g) \\
&\le \mathop{\mathbb{E}}_{(x,y)\sim\mathcal{D}}[p_g(x)(y - p(x))] - \mathop{\mathbb{E}}_{(x,y)\sim\mathcal{D}}[p_g(x)] \mathop{\mathbb{E}}_{(x,y)\sim\mathcal{D}}[y - p(x)] \\
&\le \alpha + \alpha. \tag{11}
\end{aligned}$$

From Equalized odds,

$$|\mathrm{Cov}(p, p_g | y = 1)| \le \alpha; \tag{12}$$
$$|\mathrm{Cov}(p, p_g | y = 0)| \le \alpha. \tag{13}$$

From our assumption that $y$ is unpredictable:

$$\frac{\mathrm{Cov}(y, p)}{\mu_1 (1 - \mu_1)} \le \frac{9}{10}. \tag{14}$$

Putting it all together:

$$\beta - 2\alpha \le \mathrm{Cov}(p, p_g) \le \alpha + 0.9\beta, \tag{15}$$

This means that $0.1\beta \le 3\alpha$, which is a contradiction.

$\square$

# E  Post-processing

## E.1  Analysis of the Demographic Parity Algorithm

In this section we analyze Algorithm 1. The claim below proves that the algorithm runs at most $1/\alpha^2$ update steps. On each update step, the algorithm must check if $\mathrm{Cov}(p, p_g) \le \alpha$ for every stereotype predictor $p_g$. This check does not requires any labeled samples $(x, y) \sim \mathcal{D}$, but approximating $\mathrm{Cov}(p, p_g)$ to an additive error of $\alpha$ requires $O(1/\alpha^2)$ unlabeled samples $x \sim \mathcal{D}$.

**Claim E.1.** *Algorithm 1 runs for at most $1/\alpha^2$ update steps.*

*Proof.* Let $p_0$ be the predictor in the beginning of the algorithm. Because it is bounded in $[0, 1]$ we have that $\mathrm{Var}(p_0) \le 1$. From Claim E.2 below, on each update the variance of the predictor reduces by at least $\alpha^2$. Since the variance of any random variable is positive, there could be at most $1/\alpha^2$ update steps to the algorithm. $\square$

**Claim E.2.** *Let $p : \mathcal{X} \to [0, 1]$ and let $p' : \mathcal{X} \to [0, 1]$ be the predictor after one step of the algorithm. Then $\mathrm{Var}(p') \le \mathrm{Var}(p) - \alpha^2$.*

*Proof.* Let $p_g$ be the stereotype predictor chosen for the update, and assume for a start that the algorithm above does not perform capping. That is, or every $x$, $p'(x) = p(x) - b\alpha(p_g(x))$. We write the variance

explicitly:

$$\begin{aligned}
\mathrm{Var}(p') &= \mathop{\mathbb{E}}_{x \sim \mathcal{D}}[(p'(x))^2] - \left( \mathop{\mathbb{E}}_{x \sim \mathcal{D}}[p'(x)] \right)^2 \\
&= \mathop{\mathbb{E}}_{x \sim \mathcal{D}}[(p(x) - b\alpha p_g(x))^2] - \left( \mathop{\mathbb{E}}_{x \sim \mathcal{D}}[p(x) - b\alpha(p_g(x))] \right)^2 \\
&= \mathop{\mathbb{E}}_{x \sim \mathcal{D}}[p^2(x)] - 2b\alpha \mathop{\mathbb{E}}_{x \sim \mathcal{D}}[p(x)p_g(x)] + \alpha^2 \mathop{\mathbb{E}}_{x \sim \mathcal{D}}[p_g^2(x)] - \left( \mathop{\mathbb{E}}_{x \sim \mathcal{D}}[p(x)] \right)^2 \\
&\quad + 2b\alpha \mathop{\mathbb{E}}_{x \sim \mathcal{D}}[p(x)] \mathop{\mathbb{E}}_{x \sim \mathcal{D}}[p_g(x)] - \alpha^2 \left( \mathop{\mathbb{E}}_{x \sim \mathcal{D}}[p_g(x)] \right)^2 \\
&= \mathrm{Var}(p) + \alpha^2 \mathrm{Var}(p_g) - 2b\alpha \mathrm{Cov}(p, p_g).
\end{aligned}$$

Since $b = \mathrm{sign}(\mathrm{Cov}(p, p_g))$, we get $b\mathrm{Cov}(p, p_g) = |\mathrm{Cov}(p, p_g)|$. From our assumption that $p_g$ is chosen to update, we have that $|\mathrm{Cov}(p, p_g)| \geq \alpha$. Since $p_g$ is a bounded random variable in $[0, 1]$, its variance is also bounded $\mathrm{Var}(p_g) \leq 1$. Together we have that

$$\mathrm{Var}(p') \leq \mathrm{Var}(p) - \alpha^2, \tag{16}$$

as required.

Capping the value of $p'$ to $[0, 1]$ can only reduce its variance, and therefore the inequality in (16) holds even when capping is done. □

### E.2 Analysis of the Equal Opportunity Algorithm

In this section we analyse Algorithm 2. This algorithm updates a predictor $p$ to satisfy equal opportunity with respect to a single stereotype predictor $p_g : \mathcal{X} \to R$. There is only a single update, but in order to find the updated set $R' = \{r \in R \mid r > \mu_g\}$ and update ratio $\gamma$ the algorithm must approximate many parameters.

1. $\mu_g = \mathbb{E}_{(x,y) \sim \mathcal{D}}[p_g(x)|y = 1]$ up to an additive factor of $\alpha^2/8$.

2. We define $R'$ based of our approximations of $\mu_g$. Approximate $\mathrm{Cov}(p_1, p_g|y = 1), \mathrm{Cov}(p_2, p_g|y = 1)$ up to an additive error of $\alpha^2/8$. If the approximation of $\mathrm{Cov}(p_2, p_g|y = 1)$ is positive, change it to 0.

Approximation up to an additive factor of $\alpha^2$ can be done using $O(1/\alpha^4)$ samples.

**Claim E.3.** *For every predictor $p : \mathcal{X} \to [0, 1]$ and stereotype predictor with a discrete range $p_g : \mathcal{X} \to R$ such that $\mathrm{Cov}(p, p_g|y = 1) > \alpha > 0$, Algorithm 2 outputs a predictor $p' : \mathcal{X} \to [0, 1]$ such that $\mathrm{Cov}(p, p_g|y = 1) < \alpha$.*

*Proof of Claim E.3.* Let $p, p', p_g$ be as in the claim, and assume that the algorithm approximates the values $\mu_g, \mu$ up to an additive error as described above.

Let $R' \subset R$ be the set of values chosen by the algorithm, and let $p_1, p_2 : \mathcal{X} \to [0, 1]$ be the predictors as defined in the algorithm:

$$p_1(x) = \begin{cases} p(x) & x \in R' \\ 0 & \text{otherwise.} \end{cases} \qquad p_2(x) = \begin{cases} 0 & x \in R' \\ p(x) & \text{otherwise.} \end{cases}$$

For every $x$, $p(x) = p_1(x) + p_2(x)$ and $p'(x) = \gamma p_1(x) + p_2(x)$. Since covariance is additive, we have that $\mathrm{Cov}(p', p_g|y = 1) = \gamma \mathrm{Cov}(p_1, p_g|y = 1) + \mathrm{Cov}(p_2, p_g|y = 1)$.

We bound the value of $\mathrm{Cov}(p_2, p_g | y = 1)$ for our choice of $R'$, taking into account the approximation error in $\mu_g$.

$$\mathrm{Cov}(p_2, p_g | y = 1) = \sum_{r \notin R'} \Pr_{x \sim \mathcal{D}}[p_g(x) = r | y = 1] \mathop{\mathbb{E}}_{x \sim \mathcal{D}}[p(x) | y = 1, p_g(x) = r](r - \mathop{\mathbb{E}}_{x \sim \mathcal{D}}[p_g(x) | y = 1])$$

$$\leq \frac{\alpha^2}{8} + \sum_{r \notin R'} \Pr_{x \sim \mathcal{D}}[p_g(x) = r | y = 1] \mathop{\mathbb{E}}_{x \sim \mathcal{D}}[p(x) | y = 1, p_g(x) = r](r - \mu_g)$$

$$\leq \frac{\alpha^2}{8}.$$

Where the last inequality is because of our choice of $R'$. We remark that without approximation errors in $\mu_g$, we have that $\mathrm{Cov}(p_2, p_g | y = 1) \leq 0$. The approximation error leads to a slightly weaker bound.

From the assumptions of the claim, we have that $\mathrm{Cov}(p, p_g | y = 1) = \alpha$, this implies that in fact $\mathrm{Cov}(p_1, p_g | y = 1) \geq \alpha - \alpha^2/2$. We approximated $\mathrm{Cov}(p_1, p_g | y = 1)$ up to an additive error of $\alpha^2/8$, and therefore our approximation of $\mathrm{Cov}(p_1, p_g | y = 1)$ is bounded away from 0.

From our assumption, Algorithm 2 approximates $\mathrm{Cov}(p_2, p_g | y = 1), \mathrm{Cov}(p_1, p_g | = 1)$ up to an additive error of $\alpha^2/8$. If $\gamma$ is the ratio chosen by the algorithm, this means that

$$\left| \gamma - \frac{\mathrm{Cov}(p_2, p_g | y = 1)}{\mathrm{Cov}(p_1, p_g | y = 1)} \right| \leq \alpha/2. \tag{17}$$

Combining it all, we get that

$$\mathrm{Cov}(p', p_g | y = 1) = -\gamma \mathrm{Cov}(p_1, p_g | = 1) + \mathrm{Cov}(p_2, p_g | y = 1) \leq \alpha. \tag{18}$$

We are left with showing that the transformation outputs a predictor $p' : \mathcal{X} \to [0, 1]$, i.e. that $\gamma \in [0, 1]$. From our choice of $R'$ we have that $\mathrm{Cov}(p_1, p_g | = 1) \geq 0$ always, and by definition, $\mathrm{Cov}(p_2, p_g | = 1) \leq 0$. We remark that in the case that we forced our approximation of $\mathrm{Cov}(p_2, p_g | = 1)$ to be 0, (17) still hold because of our bound that $\mathrm{Cov}(p_2, p_g | y = 1) \leq \frac{\alpha^2}{8}$. From construction and since covariance is additive we have that $\mathrm{Cov}(p, p_g | y = 1) = \mathrm{Cov}(p_1, p_g | y = 1) + \mathrm{Cov}(p_2, p_g | y = 1)$. Our assumption is that $\mathrm{Cov}(p, p_g | = 1) > \alpha$, and therefore $|\mathrm{Cov}(p_1, p_g | = 1)| > |\mathrm{Cov}(p_2, p_g | = 1)|$. This implies that $\gamma \in [0, 1]$. $\square$

### E.3 Multiaccuracy

In this section we prove Claim 6.1, showing that an $\alpha$-multiaccurate predictor satisfies covariance multiaccuracy while paying a factor of 2 in the error term. Requiring $p$ to be accurate with respect to the constant function 1 is requiring that the expected value of $p$ is the same as of $y$: $|\mathbb{E}_{(x,y) \sim \mathcal{D}}[p(x)] - \mathbb{E}_{(x,y) \sim \mathcal{D}}[y]| \leq \alpha$.

*Proof of Claim 6.1.* We write the covariance explicitly:

$$\mathrm{Cov}(p, p_g) = \mathbb{E}[p(x) p_g(x)] - \mathbb{E}[p(x)] \mathbb{E}[p_g(x)]. \tag{19}$$

The predictor $p$ is $\alpha$-accurate with respect to $p_g$:

$$|\mathbb{E}[p_g(x)(y - p(x))]| \leq \alpha.$$

We can substitute $\mathbb{E}[p_g(x) p(x)]$ in $\mathrm{Cov}(p, p_g) + \mathbb{E}[p(x)] \mathbb{E}[p_g(x)]$ it in the equation above:

$$|\mathbb{E}[y p_g(x)] - \mathrm{Cov}(p, p_g) - \mathbb{E}[p(x)] \mathbb{E}[p_g(x)]| \leq \alpha. \tag{20}$$

The predictor $p$ is also accurate with respect to the constant function 1: $|\mathbb{E}[p(x)] - \mathbb{E}[y]| \leq \alpha$. Using the triangle inequality, We can replace $\mathbb{E}[p(x)]$ by $\mathbb{E}[y]$ in (20) and pay an additional additive error of $\alpha$ (as $\mathbb{E}[p_g(x)] \in [0, 1]$):

$$|\mathbb{E}[y p_g(x)] - \mathrm{Cov}(p, p_g) - \mathbb{E}[y] \mathbb{E}[p_g(x)]| \leq 2\alpha. \tag{21}$$

We have that $\mathrm{Cov}(y, p_g) = \mathbb{E}[y p_g(x)] - \mathbb{E}[y] \mathbb{E}[p_g(x)]$, and therefore, $|\mathrm{Cov}(y, p_g) - \mathrm{Cov}(p, p_g)| \leq 2\alpha$. $\square$

### E.4   Optimal Post Processing

Here we formally define the optimal post-processing of a predictor $p$ with respect to a set of fairness constraints. In the post-processing, we want a function $f(p(x), p_g(x))$ that minimizes the loss and satisfies the constraints. Since we assumed both $p, p_g$ have a discrete range, we describe it as a convex optimization function in the possible values of post-processing.

Let $I_1 \subset I$ be the set of indices to stereotype predictors in which we want to use for demographic parity, and $I_2, I_3$ the indices of stereotype predictors for equal opportunity and multiaccuracy, respectively. We estimate for every $r \in R^{|I|+1}$ the values $\beta_r = \Pr_{x \sim \mathcal{D}}[\forall i, p_{g_i}(x) = r_i, p(x) = r_{|I|+1}]$ and $\gamma_r = \Pr_{(x,y) \sim \mathcal{D}}[\forall i, p_{g_i}(x) = r_i, p(x) = r_{|I|+1}|y = 1]$.

In order to ensure that the solution satisfies the constraints, we should approximate each $\beta_r, \gamma_r$ up to an additive factor of $\frac{\alpha}{10|R|^{|I|+1}}$. The complexity of the optimization problem and the required sample complexity for estimating the required values have an exponential dependency in the number of fairness constraints, $|I|$. We remark that we can assume $y \sim \text{Ber}(p(x))$ when estimating the expected loss, which then does not require new samples of $y$ given approximations to $\beta, \gamma, \mu$. Therefore, the optimization problem is efficient only when there are very few different fairness constraints.

$$\min \sum_{r \in R^{|S|+1}} \beta_r \mathop{\mathbb{E}}_{(x,y) \sim \mathcal{D}}[\ell(z_r, y)|\forall i, p_{g_i}(x) = r_i, p(x) = r_{|I|+1}] \tag{22}$$

$$\text{s.t. } \forall r, \quad 0 \le z_r \le 1 \tag{23}$$

$$\forall i \in I_1, \quad \left| \sum_r \beta_r r_i z_r - \left( \sum_r \beta_r z_r \right) \left( \sum_r \beta_r r_i \right) \right| \le \alpha \tag{24}$$

$$\forall i \in I_2, \quad \left| \sum_r \gamma_r r_i z_r - \left( \sum_r \gamma_r z_r \right) \left( \sum_r \gamma_r r_i \right) \right| \le \alpha, \tag{25}$$

$$\forall i \in I_3, \quad \left| \sum_r \beta_r r_i z_r - \left( \sum_r \beta_r z_r \right) \left( \sum_r \beta_r r_i \right) - \sum_r \mu \gamma_r r_i + \mu \left( \sum_r \beta_r r_i \right) \right| \le \alpha. \tag{26}$$

## F   Comparing to Existing Work Through the Lens of Covariance

### F.1   Empirical measurements of amplification

With our notion of covariance for stereotypes, we can describe stereotypes arising from amplified dataset bias described in prior work. Zhao et. al. (Zhao et al., 2017) examine stereotypical associations between gender and activity in visual semantic role labeling and multiclass classification. In their task, the same model outputs both a demographic attribute (e.g. `man, woman`) and an activity present in the image (e.g. `cooking`). Here, we see that existing notions of bias amplification can be related to the sum of differences in covariance across all groups and tasks to be measured. Subsequent definitions of bias amplification have built on this work that considers group imbalance and both positive and negative associations Wang & Russakovsky (2021).

**Definition F.1** (Bias Amplification). *The bias amplification of $p$ with respect to a stereotype predictor $p_g$ is:*

$$\delta = \text{Cov}(y, g) - \text{Cov}(p, p_g).$$

We relate bias amplification as in Definition F.1 to multiaccuracy when the stereotype predictor $p_g$ satisfies two simple accuracy conditions.

**Claim F.2.** *Let $p_g : \mathcal{X} \to [0, 1]$ be a stereotype predictor for a group membership $g$, and let $y$ be the outcome that we care about. If $p_g$ satisfies the following two conditions.*

  *1. $|\mathbb{E}_{(x,y) \sim \mathcal{D}}[p_g(x)] - \mathbb{E}_{(x,y) \sim \mathcal{D}}[g(x)]| \le \alpha$,*

2. $|\mathbb{E}_{(x,y)\sim\mathcal{D}}[p_g(x)|y=1] - \mathbb{E}_{(x,y)\sim\mathcal{D}}[g(x)|y=1]| \leq \alpha.$

*Then any predictor $p : \mathcal{X} \to [0,1]$ for $y$ that satisfies the $\alpha$-covariance multiaccuracy with respect to a set $S$ containing $p_g$, then $p$ has bias amplification at most $|\delta| = 3\alpha$ with respect to $p_g$.*

*Proof.* Let $p_g$ be a stereotype predictor satisfying the conditions of the claim, and let $p$ be a predictor that is $\alpha$-covariance multiaccuracy condition with respect to $p_g$. Then, by definition $p, p_g$ satisfy

$$|\text{Cov}(p, p_g) - \text{Cov}(y, p_g)| \leq \alpha.$$

To finish the proof we only need to bound $|\text{Cov}(y,g) - \text{Cov}(y,p_g)|$. We start from upper bounding $\text{Cov}(y,p_g)$:

$$
\begin{aligned}
\text{Cov}(y, p_g) &= \mathbb{E}_{(x,y)\sim\mathcal{D}}[y p_g(x)] - \mathbb{E}_{(x,y)\sim\mathcal{D}}[y] \, \mathbb{E}_{(x,y)\sim\mathcal{D}}[p_g(x)] \\
&= \mathbb{E}_{(x,y)\sim\mathcal{D}}[y] \left( \mathbb{E}_{(x,y)\sim\mathcal{D}}[p_g(x)|y=1] - \mathbb{E}_{(x,y)\sim\mathcal{D}}[p_g(x)] \right) \\
&\leq \mathbb{E}_{(x,y)\sim\mathcal{D}}[y] \left( \mathbb{E}_{(x,y)\sim\mathcal{D}}[g(x)|y=1] + \alpha - \mathbb{E}_{(x,y)\sim\mathcal{D}}[g(x)] + \alpha \right) \\
&\leq \text{Cov}(y,g) + 2\alpha.
\end{aligned}
$$

Where we use the fact that $p_g$ satisfies the two conditions of the claim, and that $y$ is binary. The same argument (up to sign) gives use a lower bound for $\text{Cov}(y, p_g)$:

$$
\begin{aligned}
\text{Cov}(y, p_g) &= \mathbb{E}_{(x,y)\sim\mathcal{D}}[y p_g(x)] - \mathbb{E}_{(x,y)\sim\mathcal{D}}[y] \, \mathbb{E}_{(x,y)\sim\mathcal{D}}[p_g(x)] \\
&= \mathbb{E}_{(x,y)\sim\mathcal{D}}[y] \left( \mathbb{E}_{(x,y)\sim\mathcal{D}}[p_g(x)|y=1] - \mathbb{E}_{(x,y)\sim\mathcal{D}}[p_g(x)] \right) \\
&\geq \mathbb{E}_{(x,y)\sim\mathcal{D}}[y] \left( \mathbb{E}_{(x,y)\sim\mathcal{D}}[g(x)|y=1] - \alpha - \mathbb{E}_{(x,y)\sim\mathcal{D}}[g(x)] - \alpha \right) \\
&\leq \text{Cov}(y,g) - 2\alpha.
\end{aligned}
$$

The two bounds on $\text{Cov}(y, p_g)$ implies that $|\text{Cov}(y, p_g) - \text{Cov}(y, g)| \leq 2\alpha$. Using the triangle inequality:

$$|\text{Cov}(y,g) - \text{Cov}(p, p_g)| \leq |\text{Cov}(y, p_g) - \text{Cov}(p, p_g)| + |\text{Cov}(y,g) - \text{Cov}(y, p_g)| \leq \alpha + 2\alpha,$$

which finishes the proof. $\qquad\square$

**Definition F.3.** *Bias Amplification (Zhao et al., 2017) Let $\mathcal{A}$ be the set protected demographic groups, each protected group $a \in \mathcal{A}$ has a corresponding, binary random variable $A_a$. Let $\mathcal{T}$ be the set of target tasks, with each target task $t \in \mathcal{T}$ has a corresponding binary variable $T_t$:*

$$BiasAmp = \frac{1}{|\mathcal{T}|} \sum_{a\in\mathcal{A}, t\in\mathcal{T}} y_{at} \Delta_{at}$$

$$y_{at} = \mathbf{1}[\Pr[A_a = 1|T_t = 1] > \frac{1}{|\mathcal{A}|}]$$

$$\Delta_{at} = \Pr[\hat{A}_a = 1|\hat{T}_t = 1] - \Pr[A_a = 1|T_t = 1]$$

*where $\hat{A}_a$ and $\hat{T}_t$ are model predictions.*

**Lemma F.4.** *For two protected groups (i.e. $|\mathcal{A}| = 2$) and one outcome task (i.e. $|\mathcal{T}| = 1$), bias amplification can be written in terms of covariance as follows:*

$$\Delta_{at} = \frac{\text{Cov}(\hat{A}_a, \hat{T}_t)}{\Pr[\hat{T}_t = 1]} + \Pr[\hat{A}_a = 1] - \frac{\text{Cov}(A_a, T_t)}{\Pr[T_t = 1]} - \Pr[A_a = 1]$$

*Proof.* For binary variables $A$ and $T$, we can write:

$$\text{Cov}(A, T) = \mathbb{E}[AT] - \mathbb{E}[A]\,\mathbb{E}[T]$$
$$= \Pr[A = 1, T = 1] - \Pr[A = 1]\Pr[T = 1]$$
$$\Pr[A = 1|T = 1] = \frac{\text{Cov}(A, T)}{\Pr[T = 1]} + \Pr[A = 1]$$

Since $A$, $T$, $\hat{A}$ and $\hat{T}$ are binary random variables:

$$\Delta_{at} = \Pr[\hat{A}_a = 1|\hat{T}_t = 1] - \Pr[A_a = 1|T_t = 1]$$
$$= \frac{\text{Cov}(\hat{A}, \hat{T})}{\Pr[\hat{T} = 1]} + \Pr[\hat{A} = 1] - \frac{\text{Cov}(A, T)}{\Pr[T = 1]} + \Pr[A = 1]$$

$\square$

Here, we see that existing notions of bias amplification can be related to the sum of differences in covariance across all groups and tasks to be measured.

However, Definition F.5 suffers from several shortcomings:

- Only positive associations are included (i.e. $\Pr[\hat{A}_a = 1|\hat{T}_t = 0]$ is not considered)

- $y_{at}$ does not consider group imbalance (i.e. if $\Pr[A_a = 1|T_t = 1] < \frac{1}{|\mathcal{A}|}$ for some group $a \in A$, amplification is ignored)

- Amplification is summarized across tasks, individual tasks and protected groups may experience amplification in different directions while overall BiasAmp remains 0

To address the first two shortcomings and provide disambiguation for whether amplification arises from $\hat{A}$ or $\hat{T}$, (Wang & Russakovsky, 2021) provide a directional version of Definition F.5:

**Definition F.5.** *Directional Bias Amplification (Wang & Russakovsky, 2021) Let $\mathcal{A}$ be the set protected demographic groups, each protected group $a \in \mathcal{A}$ has a corresponding, binary random variable $A_a$. Let $\mathcal{T}$ be the set of target tasks, with each target task $t \in \mathcal{T}$ has a corresponding binary variable $T_t$:*

$$BiasAmp_{\rightarrow} = \frac{1}{|\mathcal{T}||\mathcal{A}|} \sum_{a \in \mathcal{A}, t \in \mathcal{T}} y_{at}\Delta_{at} + (1 - y_{at})(-\Delta_{at})$$
$$y_{at} = 1[\Pr[A_a = 1, T_t = 1] > \Pr[A_a = 1]\Pr[T_t = 1]]$$
$$\Delta_{at} = \begin{cases} \Pr[\hat{T}_t = 1|\hat{A}_a = 1] - \Pr[T_t = 1|A_a = 1] & \text{if measuring } A \rightarrow T \\ \Pr[\hat{A}_a = 1|\hat{T}_t = 1] - \Pr[A_a = 1|T_t = 1] & \text{if measuring } T \rightarrow A \end{cases}$$

*where $\hat{A}_a$ and $\hat{T}_t$ are model predictions.*

**Corollary F.6.** *For two protected groups (i.e. $|\mathcal{A}| = 2$) and one outcome task (i.e. $|\mathcal{T}| = 1$), bias amplification can be written in terms of covariance as follows:*

$$\Delta_{at} = \begin{cases} \frac{\text{Cov}(\hat{A}_a, \hat{T}_t)}{\Pr[\hat{T}_t = 1]} + \Pr[\hat{A}_a = 1)] - \frac{\text{Cov}(A_a, T_t)}{\Pr[T_t = 1]} - \Pr[A_a = 1] \\ \frac{\text{Cov}(\hat{A}_a, \hat{T}_t)}{\Pr[\hat{A}_a = 1]} + \Pr[\hat{T}_t = 1] - \frac{\text{Cov}(A_a, T_t)}{\Pr[A_a = 1]} - \Pr[T_t = 1] \end{cases}$$
$$y_{at} = 1[\text{Cov}(A_a, T_t) > 0]$$

*Proof.* The proof is similar to the prior proof using the identity for binary random variables $A$, $T$, $\hat{A}$ and $\hat{T}$:

$$\Pr[A = 1|T = 1] = \frac{\text{Cov}(A, T)}{\Pr[T = 1]} + \Pr[A = 1]$$

For $y_{at}$, we can see that is exactly the definition of covariance for binary variables. $\square$

