# OpenReview forum: "Fairness with respect to Stereotype Predictors: Impossibilities and Best Practices"
_TMLR — Accepted by TMLR_

### Review · Reviewer_SNpi · 2024-12-18

**Summary Of Contributions:**

This paper adapts notions of fairness (viz. demographic parity, equal
opportunity, multi-accuracy) wrt. groups  to fairness wrt. to stereotypes (viz.
stereotypical associations, harmful stereotypes, and prototypical vs. outlier
individuals).  These formulations of fairness wrt. stereotypes are formally
defined, and the authors prove certain bounds regarding simultaneously
achieving fairness and task performance (i.e., accuracy for
classifiers/predictor).  Post-processing algorithms are then given to show how
a given classifier (predictor) can be modified _post hoc_ to conform to various
notions of accuracy (possibly at the expense of task performance).  Finally,
the authors present two empirical case studies to demonstrate the application
of the proposed formulations and post-processing algorithms.

**Audience:**

Yes

**Broader Impact Concerns:**

I think the limitations section is a little light.  For example, are there any
ways that the findings of the paper could be used maliciously, such as for
finding predictors which satisfy DP/EO/MA for trivially selected stereotypes
that nevertheless seem convincing to the careless eye?  This paper might not
enable it _per se_, but the framework and algorithms could make it easier to do
so.  The answer to this might be "no" (I am not well-versed in this particular
field), but it is not apparent that these deeper limitations are considered.
In either case, I do not see any limitations or broader impact concerns that
would be highly problematic for this paper, but a more robust discussion would
be good.

**Claims And Evidence:**

Yes

**Requested Changes:**

### Contextualizing post-processing algorithms
One addition or edit I think would be good (though not critical) would be to
contextualize the post-processing algorithms for Sections 6.1 and 6.2.  Namely,
what is good about these PP algorithms that would make them better than, say,
something trivial like reducing the predictor to a constant/mean of the
population?  The empirical case studies show that the PP algorithms are
obviously better than this trivial case, but I think it would be good to either
give a proof or even just an intuition as to why these PP algorithms are good
even if not optimal (e.g., this is halfway done with the fact that they are
tractable (cf. optimal PP)).

### Minor changes
- [Sec 1] fix quotes around "stereotypic thinking"
- [Sec 2] space after "occurs within a group"
- [Sec 4.1] What notion of correlation is being discussed in the last
  paragraph?  Linear (Pearson) or rank (e.g., Spearman)?
- [Sec 6.4] "minimazes" -> "minimizes"
- [Sec 7] I did not have a great sense of what a high vs. low covariance was,
  so if there were some way to put those values in perspective briefly, it
  could be helpful.
- [Sec 8] "but less sweeping the impossibility" -> "but less sweeping than the
  impossibility"

**Strengths And Weaknesses:**

My assessment of the strengths and weaknesses of the paper should be prefaced
with the fact that I am not intimately familiar with body of literature on
algorithmic fairness.  I can understand the definitions and most of the claims
formally, but I was not able to verify the proofs (even if I could follow some
of them).  Similarly, if there is existing work that overlaps significantly
with this, I would not be aware of it.  Thus, my assessment will stay on
a primarily conceptual level.

### Strengths
- (major) This paper is cohesive and complete offering theoretical accounts,
  proofs, empirical analysis, and framing of the problem statement.  It offers
  some of the bird's-eye perspective of a survey paper alongside its more
  concrete contributions.
- (major) I think the contextualization of the paper and its findings are good.
  It is easy to see how the notions of fairness wrt. stereotypes are related
  but not trivially to prior work on fairness wrt. groups.  The findings
  regarding limitations on accuracy vis-a-vis demographic parity and equal
  opportunity are especially relevant as they head off the naive suggestions of
  "why can't we just make all predictors unbiased in every way possible?".
- (minor) The empirical case studies round out the paper well, demonstrating
  the formally discussed principles in action.  The discussion of the empirical
  results is somewhat brief, so it is not clear how significant the findings
  are (e.g., vs. comparable methods), but I understand space in the paper is
  limited.

### Weaknesses
- (minor) See statement in _Requested Changes_ about "Contextualizing post-processing algorithms".

---

> ### Author Response · Authors · 2025-03-17
> **Thank you for your review! Overview of Changes:**
>
> We’d like to thank the reviewer for highlighting the strengths of our paper and giving us guidance on how to improve our paper further. We have incorporated the changes suggested by the reviewer in blue text in this second version of the draft.
>
> An overview of our changes is as follows:
> 1. **An additional explanation of post-processing algorithms** :
> The post-processing algorithms are meant to minimize the changes to the predictor while satisfying the fairness constraints. While setting the predictor to a constant function would satisfy the fairness constraint, it would not be a helpful predictor. In Appendix E.1, we added a claim and a proof for bounding the difference between p (original predictor) and the post-processed predictor. We also slightly modified our algorithm to further reduce the distance between the original and post-processed predictor.
> Enforcing a large number of fairness constraints with respect to different stereotypes might force the post-processed predictor to be constant. In these cases, our post-processing algorithm will also output a constant predictor, however, this is not the case for the stereotypes in our experiments.
> 2. **Expanding on Limitations**: The reviewer brings up a great point that it is indeed possible to use our framework with maliciously defined stereotypes to achieve unwanted goals. In real life, even a benevolent party enforcing fairness with respect to stereotypes in real-life settings could encounter unintended adverse impacts. Thus, practitioners should be careful, as we have demonstrated, with the impacts of interventions on accuracy and fairness with respect to other members of the minority group. We added additional discussion in the limitation section to discuss the importance of using fairness with respect to stereotypes only with very carefully chosen stereotypes, as well as potential misuse.
>
> We hope we have adequately addressed questions that the reviewer may have. We are happy to make additional changes or answer additional questions.

---

### Review · Reviewer_sXEC · 2024-12-19

**Summary Of Contributions:**

The submission investigates algorithmic fairness, specifically addressing representational harms such as stereotypes in machine learning models. It introduces a unifying framework called "stereotype predictors", which consolidates various fairness notions and connects them to existing group fairness metrics. The authors provide theoretical guidance for selecting stereotype predictors, propose algorithms to mitigate stereotypes under different fairness notions, and validate their approach through two empirical case studies. This work aims to offer practical tools for tackling representational harms in downstream AI systems, making a contribution to the algorithmic fairness literature.

**Audience:**

Yes

**Broader Impact Concerns:**

No comments.

**Claims And Evidence:**

Yes

**Requested Changes:**

The mentioned weaknesses.

**Strengths And Weaknesses:**

Strengths:
1. Well-Defined Framework: The introduction of stereotype predictors as a unifying concept is innovative and aligns with ongoing discussions in fairness research.
2. Theoretical Contributions: The mapping of stereotype predictors to group fairness notions is grounded in theory and provides insights for practitioners.
3. Empirical Validation: The inclusion of two case studies strengthens the submission by demonstrating the feasibility and impact of the proposed methods.
4. Relevance to Fairness Research: Tackling representational harms is crucial for achieving equitable AI systems, and the submission addresses this need effectively.

Weaknesses:
1. Structural Omissions: The absence of a “Concluding Remarks” section is notable, as such a section would help reinforce the paper’s contributions and implications.
2. Clarify how the proposed framework relates to existing works that utilise covariances to achieve or evaluate fairness. Covariances have been employed in both linear and kernel methods for fairness. Is the proposed framework general enough to include these approaches?
3. Presentation: Minor typographical issues, such as missing spaces before parenthetical citations (pages 3 and 9).
4. Do not mention whether source code is available.

---

> ### Author Response · Authors · 2025-03-17
> **Thank you for the feedback, changes incorporated**
>
> We appreciate the reviewer for taking the time to read our paper and providing valuable feedback. We think these changes will strengthen our work, and we have incorporated your suggestions in this updated version V2 (please find all 4 proposed changes in blue text when applicable). Below is an overview of the changes we made:
>
> 1. **Concluding remarks**: We have added a section at the end of our work summarizing our work and the perspective we hope future work introducing new notions of stereotypes will consider.
> 2. **Source code**: We plan to make source code available with the release of our paper so that all of our post-processing algorithms are easy to replicate.
> 3. **Presentation**: Thank you for the feedback about the formatting. We have fixed all the changes in this updated version.
> 4. **Prior work**: Other work that use notions adjacent to covariance include Mary et al., 2019, and Grari et al., 2019 where they use maximal correlation.  These works use Hirschfeld-Gebelin-Renyi maximum correlation (equivalent to independence) as a regularizer during training. Our work considers post-processing approaches and weaker notions than independence. We have updated this clarification in this updated version (V2). As for kernel learning, we are not exactly sure what the reviewer meant. Are you referring to works such as “Fair Kernel Learning” and how it relates to our framework? We are happy to clarify further.
>
> We appreciate the review and your engagement with the process of helping make our paper better! Let me know if we can provide any additional clarifications.

---

### Review · Reviewer_vGGw · 2025-03-10

**Summary Of Contributions:**

This is an interesting paper that looks at stereotype properties and tries to model this concern via definitions of fairness.

**Audience:**

Yes

**Broader Impact Concerns:**

In general, I think from a mathematical point of view, this seems reasonable. But beyond that, I find this investigation and this characterization deeply problematic. The idea that you can yet again model some fundamental societal or human trait using a quantified notion and then justify definitions of fairness concerning this notion seems very strange and worrying. I admit this is somewhat of a vague philosophical point, but I do find it concerning.

Towards the end of the paper, they provide some experimental results showing how this definition works in a case study dataset. But I would have still liked to see some sort of evaluation with human evaluators on how well this notion captures their concerns about being treated fairly.

**Claims And Evidence:**

Yes

**Requested Changes:**

See below

**Strengths And Weaknesses:**

Primarily, the idea is to represent a function that captures the distribution of the stereotypes in the model and then characterize different types of fairness, such as demographic parity, using the stereotype predictor as a covariance function.
The treatment of this function straightforward but precise and rigorous.

---

> ### Author Response · Authors · 2025-03-17
> **The Focus of Our Work: Interrogating Computational Stereotypes**
>
> Thank you for reading our paper and providing feedback. The main concern of the reviewer lies in the general idea of fairness with respect to stereotypes, and that perceptions of stereotypes as individuals or as a society are a societal concept that should not be captured in an algorithm or definition. We are happy to add additional clarifications.
>
> ## Stereotypes as a Complex Social Construct
>
> We agree with the reviewer that the notion of a stereotype is a complex social phenomenon. However, many works have studied the measurement of stereotypes in machine learning models (Cryan et al., SanchezJunquera et al., Seaborn et al., referenced in our paper) and presented compelling evidence of the harms that may result. When identifying harms, it is no longer about the human trait of how machine behavior reinforces existing biases.
>
> Because there has been so much work studying stereotypes in machine learning, they cover a confusing spectrum of concepts. Our work is unique in creating a general framework for better understanding fairness with respect to different notions of stereotypes and does not explicitly prescribe any specific notions of stereotypes. We also show the risks of requiring some of the standard group fairness definitions, such as demographic parity and equal opportunity, with respect to different notions of stereotypes. One of our results is that requiring these fairness definitions with respect to stereotypes that are too correlated with the outcome has a price in terms of accuracy.  Our work uniquely addresses the possibility that some notions of stereotypes might not be suitable for a specific scenario by considering many different potential stereotype predictors. Our two case studies illustrate this by using three different potential stereotype predictors.
>
> We believe that the choice of which fairness definition to use with respect to stereotypes is complex and multidisciplinary and should be selected by domain experts. Our paper provides a better understanding of the implications of requiring different fairness notions with respect to different stereotypes to better assist decision-makers. The goal of the experiment section is to demonstrate that different stereotype predictors have different effects on the accuracy in post-processing algorithms. We do not suggest using these stereotypes for real-world applications, as it is not the focus of this work.
>
> ## Interrogating How Stereotypes are Operationalized
> For human studies on stereotype predictions, we highlight existing work from [1] where survey participants find that stereotype-reinforcing errors of machine learning models induce experientially harmful experiences. Other works in psychology measuring the degree and impact of stereotypes are referenced in our work as well.  However, our work serves to propose a framework for questioning and examining different notions of stereotypes through a computational lens in the context of proposed remedies rather than to test the human perceptions of stereotypes. It is out of the scope of our work to survey individuals for their thoughts on specific stereotypes. Crucially, the same notion of stereotypes that survey subjects may agree on can be operationalized in many different ways (Figure 1), thus, the computational question of evaluating an existing stereotype predictor becomes highly relevant.
>
> ## References
>
> [1] Wang, A., Bai, X., Barocas, S., & Blodgett, S. L. (2024). Measuring machine learning harms from stereotypes: requires understanding who is being harmed by which errors in what ways. arXiv preprint arXiv:2402.04420.

---

### Author Response · Authors · 2025-03-17
**We thank the reviewers for feedback, open to further questions, clarifications and edits**

Dear Reviewers,
We thank you all for reading our paper and raising suggestions and questions. Thank you for recognizing the strengths of our work.
We have uploaded a new version of our paper with changes outlined in blue.
We are happy to answer any additional questions.

---

### Author Response · Authors · 2025-03-24

As the review period comes to an end, we are happy to clarify any other questions further or incorporate any further suggestions from the reviewers.
As far as we are aware, the reviews saw many strengths of our work and we addressed all clarifications and weaknesses identified. Thanks again to all the reviewers for their engagement with this process.

---

### Decision · Action_Editor_Bn1K · 2025-04-25

**Recommendation:** Accept as is

**Comment:**

The paper generalizes several fairness definitions to use a stereotype predictor to assess the degree by which an individual belongs to a (sensitive) group. It then studies how this stereotype predictor influences the trade-off between accuracy and the generalized notions of fairness, and extend a well-known impossibility result in a straightforward manner.

The three reviewers were unfortunately not very familiar with the fairness literature and, as a result, I did independently read the paper after reading the reviews. In my opinion, the current paper does not add much with respect to related work in the fairness literature, which I am very familiar with since I have published several papers in the area, both at a conceptual level. In particular, the authors do not sufficiently motivate why incorporating stereotype predictors is reasonable. At a technical level, the paper seems sound.

**Audience:**

I have reservations that the paper will be of interest to some individuals in the algorithmic fairness community. The paper essentially generalizes slightly the definition of several fairness notions. More specifically, it considers that a stereotype predictor is used to assess the degree by which an individual belongs to a (sensitive) group.

**Claims And Evidence:**

Yes, the claims made in the submission are supported by accurate, convincing and clear evidence.